# In vivo single-cell RNA metabolic labeling resolves early transcriptional responders in the regenerating zebrafish heart

Janita Mintcheva[1,2], Tzu-Lun Tseng [3], Pinelopi Goumenaki[3], Anika Neuschulz [1,4], Anis Senoussi [1], Sara Lelek[5], Khai Lone Lim [6], Zhitao Ming[7], Ronny Schäfer[1], Alisa Hnatiuk[8,9], Nikolay Ninov [8,9], Arica Beisaw [10,11,12], Maura McGrail[7], Shih-Lei Ben Lai [6], Daniela Panáková [5,13,14], Didier Y. R. Stainier [3,15,16] & Jan Philipp Junker [1,17,18] ✉

The adult zebrafish heart can regenerate after injury, but the earliest gene expression changes that trigger this process remain poorly understood. Here we show that in vivo single-cell RNA metabolic labeling, which marks newly made RNA in individual cells, can capture rapid responses in the adult zebrafish heart after injury. Within the first 6 h, we detect activation of innate immune programs, including Toll-like receptor signaling, in a subset of macrophage-like immune cells. Analysis of a larger single-cell dataset indicates that neutrophils also contribute to this early response. Guided by these data, we show that macrophage-specific inhibition of the Toll-like receptor adaptor MyD88 reduces the pro-inflammatory macrophage response at the injury site and improves early hallmarks of regeneration. Our work establishes RNA metabolic labeling as a useful approach for measuring acute responses in vivo at single-cell resolution and identifies early immune-cell activation as a tunable component of heart regeneration.

In adult mammals, myocardial infarction leads to irreversible loss of heart tissue, as the affected tissue is replaced by a fibrotic scar[1,2]. By contrast, neonatal mice as well as adult axolotl and zebrafish possess the ability to efficiently regenerate their heart after injury[2–6]. Extensive research in zebrafish has shown that heart regeneration is a tightly orchestrated process involving multiple successive steps, including early inflammation, revascularization, formation of a regenerative scaffold and deposition of a fibrotic scar, as well as later replacement of the scar by proliferating cardiomyocytes to form functioning tissue[7]. While the processes of cardiomyocyte de-differentiation and proliferation have been studied in depth, a substantial body of recent work has also focused on the composition and contribution of the regenerative niche[8–16].

Transcriptional cell state changes are required to drive the dynamic events that underlie successful heart regeneration. For instance, cardiomyocytes acquire a de-differentiated state in order to proliferate, and fibroblasts assume a transient pro-regenerative state characterized by expression of col12a1a[8,17]. Macrophage polarization is another dynamic process during heart regeneration, where macrophages tend to be more pro-inflammatory at earlier stages, and transition to a more pro-regenerative state at later stages[18,19]. Cell state changes have also been reported in the epicardium and endocardium[9,20–22]. However, the very early response to heart injury, triggering the later regenerative cascade, remain largely unknown. While it has been reported that neutrophil and macrophage infiltration start as early as 3 h post injury, the earliest transcriptional state transitions have not been determined to date[23].

Cell fate decisions over long periods of time are typically assessed with methods like Cre-lox lineage tracing, or more recently with CRISPR/Cas9-based high-throughput lineage tracing[24–26]. However,

---

such approaches are not well-suited for measuring very dynamic and potentially short-lived transcriptional cell state changes. Specifically, measuring the response to perturbation is challenging: Analysis shortly after injury suffers from low signal, and pre-existing differences between samples can easily dominate the results. Analysis at later time points may also include secondary effects, rendering identification of direct targets difficult or impossible. We reasoned that methods for metabolic RNA labeling in single cells hold great potential for detecting short-term transcriptional state changes by allowing measurement of two time points (before and after perturbation) for each gene and each cell. However, such approaches have so far mostly been demonstrated to work in cell culture[27–32].

Here, we established the single-cell Thiol(SH)-Linked Alkylation for Metabolic Sequencing of RNA (scSLAM-seq[30]) method for RNA metabolic labeling in the adult zebrafish heart, and we used this approach to measure early transcriptional state changes upon cryoinjury. We identified macrophage-like cells as one of the earliest responders, and they upregulate innate immune response pathways such as Toll-like receptor (TLR) signaling. Perturbation of the TLR pathway, using a transgenic line with an inducible, macrophage-specific overexpression of a dominant-negative myeloid differentiation factor 88 (MyD88), led to an increase in coronary endothelial cell proliferation and coronary vessel coverage, as well as an increase in cardiomyocyte proliferation. Taken together, our data indicate that the early pro-inflammatory response needs to be balanced in terms of intensity and timing, and that the early regenerative cascade can be modulated by fine-tuning it.

## Results

### SLAM-seq labels transcription in the adult zebrafish heart

In order to measure short-term transcriptional state changes, we sought to establish metabolic labeling of RNA in the live adult zebrafish heart. Specifically, we decided on the Thiol(SH)-Linked Alkylation for

Metabolic Sequencing of RNA (SLAM-seq) method, which is based on the incorporation of 4-thiouridine (4sU) into newly transcribed RNA (Fig. 1a)[30]. After a thiol modification step using iodoacetamide, labeled transcripts can be detected in sequencing data by characteristic T-to-C conversions ("Methods"). However, several parameters like 4sU delivery, concentration, and labeling time have to be carefully optimized to avoid over- or under-labeling of transcripts. To establish our experimental setup for labeling of transcription in the adult zebrafish heart on the bulk level, we performed intrathoracic (IT) injections of different 4sU concentrations (Supplementary Data 1). While intraperitoneal (IP) injections are more common for compound delivery, we reasoned that 4sU injection in the direct proximity of the heart might reduce variability compared to IP injections. We found that IT injection of 5 µl of 200 mM 4sU showed the highest conversion rates (i.e., T-to-C substitutions per covered T position in aligned reads) in the heart without producing any adverse reaction after 6 h of labeling (Fig. 1b). To control for potential 4sU leakage in injured and sham injured hearts of IT injected zebrafish, we first IT injected 4sU and performed sham or cryoinjury 1 h after injection. We then sampled hearts at 6 h post sham or cryoinjury. We found that opening of the body wall and pericardium to perform cryoinjury did not affect conversion rates (Fig. 1b).

We next determined the 4sU conversion rate as a function of the labeling time (Supplementary Data 1). We found that after a single injection of 4sU, conversion rates increased until 12 h post injection but slightly decreased at 24 h post injection (Fig. 1c). The decrease is likely due to exhausted 4sU availability at that time. Here, we focus on shorter labeling times (typically 7 h), which is in the range of average mRNA half-life, and therefore a single injection of 4sU is sufficient. We detected relatively low overall conversion rates of approximately 1%, an approximately 3–4-fold increase in detected T-to-C conversions compared to background levels[33]. We hypothesized that this might be due to relatively low mRNA turnover in the uninjured adult heart, and that the detected labeling efficiency would be lowered by highly

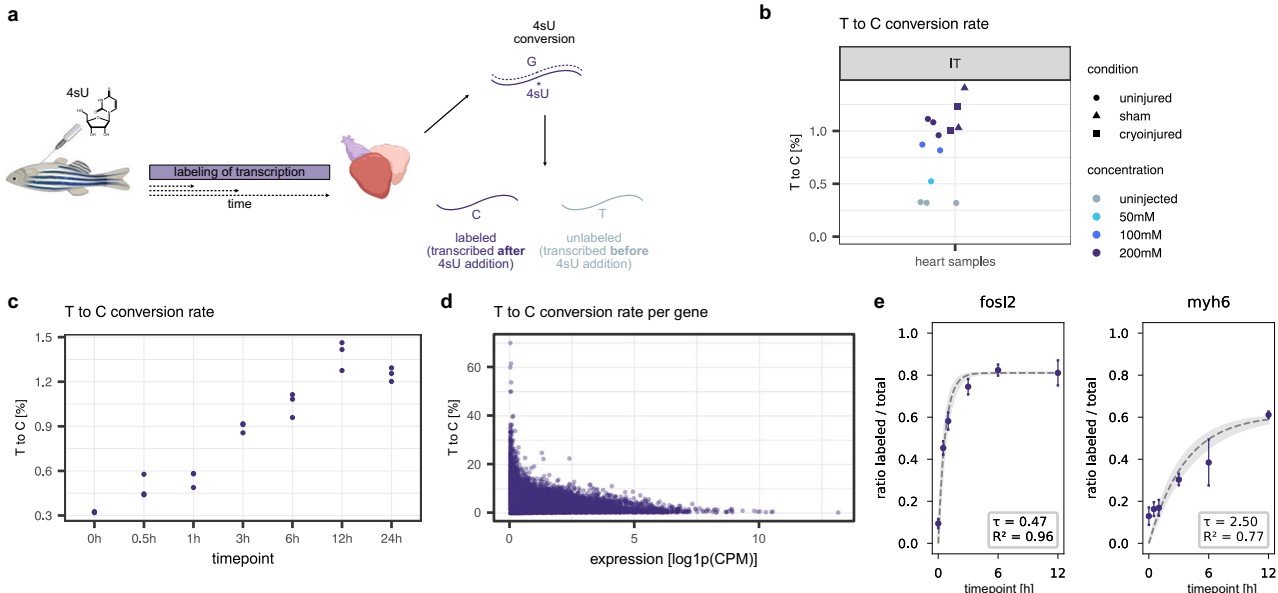

**Fig. 1 | SLAM-seq labels transcription in the adult zebrafish heart. a** Schematic bulk SLAM-seq experimental workflow. **b** T-to-C conversion rates of heart samples from IT injected zebrafish with different 4sU concentrations. Each dot represents one sample, $n = 3$ (uninjected), $n = 1$ (50 mM), $n = 2$ (100 mM), $n = 3$ (200 mM, uninjured), $n = 2$ (200 mM, sham), $n = 2$ (200 mM, cryoinjured). Data of uninjected samples was acquired in the context of the SLAM-seq time course in panel c and re-used for visualization purposes. **c** T-to-C conversion rates of SLAM-seq time course after a single 4sU injection. Each dot represents one sample, with each time point being acquired in biological triplicates. **d** T-to-C conversion rate per gene as a

function of expression after 6 h of labeling (sample 1). CPM...counts per million, log1p...natural logarithm of 1 plus the input value. **e** Two representative genes (left: *fosl2*, right: *myh6*) from the SLAM-seq time course. Dots indicate average labeled to total reads ratio among biological triplicates, error bars indicate standard deviation, dashed line indicates fitted curve, gray shaded area represents 95% confidence interval, τ values are the estimated RNA half-lives in hours. Panel (**a**) was partially created in BioRender. Mintcheva, J. (2026) https://BioRender.com/mchf1ny. For more details, see *Graphics* section ("Methods").

expressed housekeeping genes with long half-lives. Indeed, resolving the data by genes, we found a clear dependence of conversion rate on expression level, and we observed many genes with higher conversion rates (Fig. 1d).

Since SLAM-seq is based on detection of T-to-C conversions, sequencing errors and single-nucleotide variants (SNVs) can in principle lead to false positives[34]. While the effect of sequencing errors can be minimized by setting strict quality cut-offs on the reads ("Methods"), it is important to assess the possible influence of SNVs. We reasoned that SNVs would lead to very high T-to-C conversion rates for individual nucleotide positions, and we filtered against such events ("Methods"). We found that conversion rates were moderately affected, suggesting that the effect of SNVs has to be carefully evaluated (Supplementary Fig. 1a–c). By contrast, requiring at least two T-to-C conversions instead of one to classify a transcript as labeled led to a stronger decrease in the fraction of transcripts labeled (Supplementary Fig. 1c).

As both 4sU exposure and IT injections can elicit a transcriptional response, we sought to explore the effects of 4sU injection. Analysis of the time course data showed a separation of uninjured hearts labeled for 3 h and longer compared to uninjected samples in PCA space, indicating a potential underlying transcriptional response (Supplementary Fig. 1d). Moreover, differential gene expression of uninjected hearts and hearts sampled after 6 h of labeling with 4sU showed an upregulation of immune-related genes like *cxcl18*, *lect2l*, *irf1b*, *mmp9*, and *mmp13a.1*, indicating a potential inflammatory response (Supplementary Fig. 1e). However, PCA together with IP injected samples and IT injection followed by sham or cryoinjured samples showed that the transcriptional effects from sham and cryoinjury exceed those of just the 4sU injection, marked by a larger separation in PCA space (Supplementary Fig. 1f).

After these validations of the labeling strategy in the zebrafish heart, we probed if our bulk SLAM-seq data would also enable calculation of RNA turnover. We therefore quantified the ratio of labeled transcripts over total transcripts until 12 h post injection and used a mathematical model to compute decay rates (Mathematical Supplement). We identified more than 3100 genes for which we could estimate half-lives with $R^2 > 0.7$ (Fig. 1e, Supplementary Data 2). The average half-life amounted to 2.5 h ($\tau[0.5$ h$,48$ h$]$, $R^2 > 0.7$), slightly shorter than reported average half-lives in mammalian cells[35,36]. More detailed comparison with published compendia of decay rates inferred with metabolic labeling techniques showed that our calculated half-lives generally fall within the range of existing half-life compendia, taking into account the established biological and methodological variability of inferred RNA decay rates (Supplementary Fig. 2)[37].

Taken together, these data showed that transcription in adult zebrafish hearts can be quantified with SLAM-seq, making it well-suited for measuring short-term transcriptional changes such as early injury response, as the time scale of detection of labeling corresponds to the time scale of transcriptomic changes.

## scSLAM-seq resolves the early injury response in the heart

After validating that newly transcribed RNA can be robustly labeled with SLAM-seq, we next wanted to investigate if this approach could be leveraged to disentangle the early response to heart cryoinjury in the adult zebrafish in vivo. Given our estimation of RNA half-lives, we would expect that at 6 h post cryoinjury (hpci), there would still be a strong contribution from transcripts that were present prior to injury. We reasoned that the ability of scSLAM-seq to distinguish those from RNAs transcribed after injury would increase the signal-to-noise ratio and remove pre-existing inter-individual variation. We therefore adapted our experimental setup to the single cell level ("Methods") and focused on the early response within the first 6 h post cryoinjury: mRNA molecules transcribed after injury would be labeled, while mRNA present in the heart before injury would remain unlabeled.

To perform scSLAM-seq, we IT injected 4sU, and sham or cryoinjured the hearts one hour after injection. The sham control is crucial, since we cannot rule out that IT injection itself may also trigger a (possibly weaker) injury-like response. Hearts were sampled at 6 h post sham (hps) or 6 hpci, respectively, dissociated, and cells were subjected to our scSLAM-seq 10X protocol (Fig. 2a, Supplementary Data 1, see "Methods")[32,33]. The data quality of our scSLAM-seq samples was comparable to scRNA-seq data, with similar average UMI counts per cell and comparable number of genes detected per cell, though towards the lower end on the distribution range (Supplementary Data 3). We found that clustering based on the sum of labeled and unlabeled mRNA resolved all major cell types, confirming the quality of the data (Fig. 2b, c). Clustering based on only the labeled or unlabeled transcripts led to concordant results, suggesting a sufficient amount of transcriptomic information for both time points (Supplementary Fig. 3a). Furthermore, we were able to detect labeling across all cell types of the heart (Fig. 2d, Supplementary Fig. 3b), indicating that 4sU efficiently diffuses across the entire heart. This observation is in line with bulk SLAM-seq of IP injected zebrafish, as well as previous findings that after injection, labeling could be detected in many different organs of the zebrafish, suggesting that 4sU might even pass the blood–brain barrier (Supplementary Fig. 3c)[32]. We further observed approximately 20–25% of the transcripts being labeled, i.e., containing at least one T-to-C conversion, which is in line with previous studies that used SLAM-seq to measure perturbation response in cell culture (Supplementary Fig. 3d)[28]. A simplified estimation approximates the false negative rate around roughly 15%, given our observed conversion rates and an average of 50 thymines per read (Mathematical Supplement).

Next, we set out to develop a mathematical model for determining the injury response. Our model makes use of the identical experimental setup of the sham and cryoinjured sample (i.e., 4sU injection at steady state and identical labeling time) and works under the simplifying assumption that decay rates do not change in response to injury (Mathematical Supplement). In this model, the injury response is calculated as the ratio of labeled RNA at 6 hpci over the labeled RNA at 6 hps (Fig. 2e, Mathematical Supplement). Given the assumptions of our model, our measure of the injury response is not dependent on mRNA decay rates and is also independent of pre-existing unlabeled mRNA. While our model is similar to a previously published approach, we further expanded it by the addition of a background model, which takes scSLAM-seq specific noise properties into account (Mathematical supplement)[38]. Our analysis showed that macrophage-like cells displayed a specific response by upregulating KEGG terms related to pro-inflammatory innate immune response pathways like the Toll-like receptor, C-type lectin receptor (CLR), and NOD-like receptor (NLR) signaling pathways (Fig. 2f left panel, Supplementary Fig. 4a, b, Supplementary Data 4)[39]. While infiltration or activation of myeloid cells has been reported previously at later time points, we here provide direct evidence for de novo transcriptional activation within the first 6 h after cryoinjury by using our new RNA readout[10,18]. Across all cell types, we observed a generic response, which may reflect reaction to cellular stress. This response may also partially arise from ambient RNA contamination from dead cardiomyocytes, which are generally the most fragile cell type to heart dissociation (Supplementary Data 4). Interestingly, we were not able to detect this specific response in macrophage-like cells when running our model on the same data without SLAM information, or when performing standard differential gene expression analysis on the transcriptome (Fig. 2f right panel, Supplementary Data 5). This point highlights that RNA labeling information increased the sensitivity to identify injury responsive genes after perturbation. We further validated the robustness of our findings by confirming that these results were not influenced by false positives caused, e.g., by SNVs or sequencing artifacts (Supplementary Fig. 5a–c).

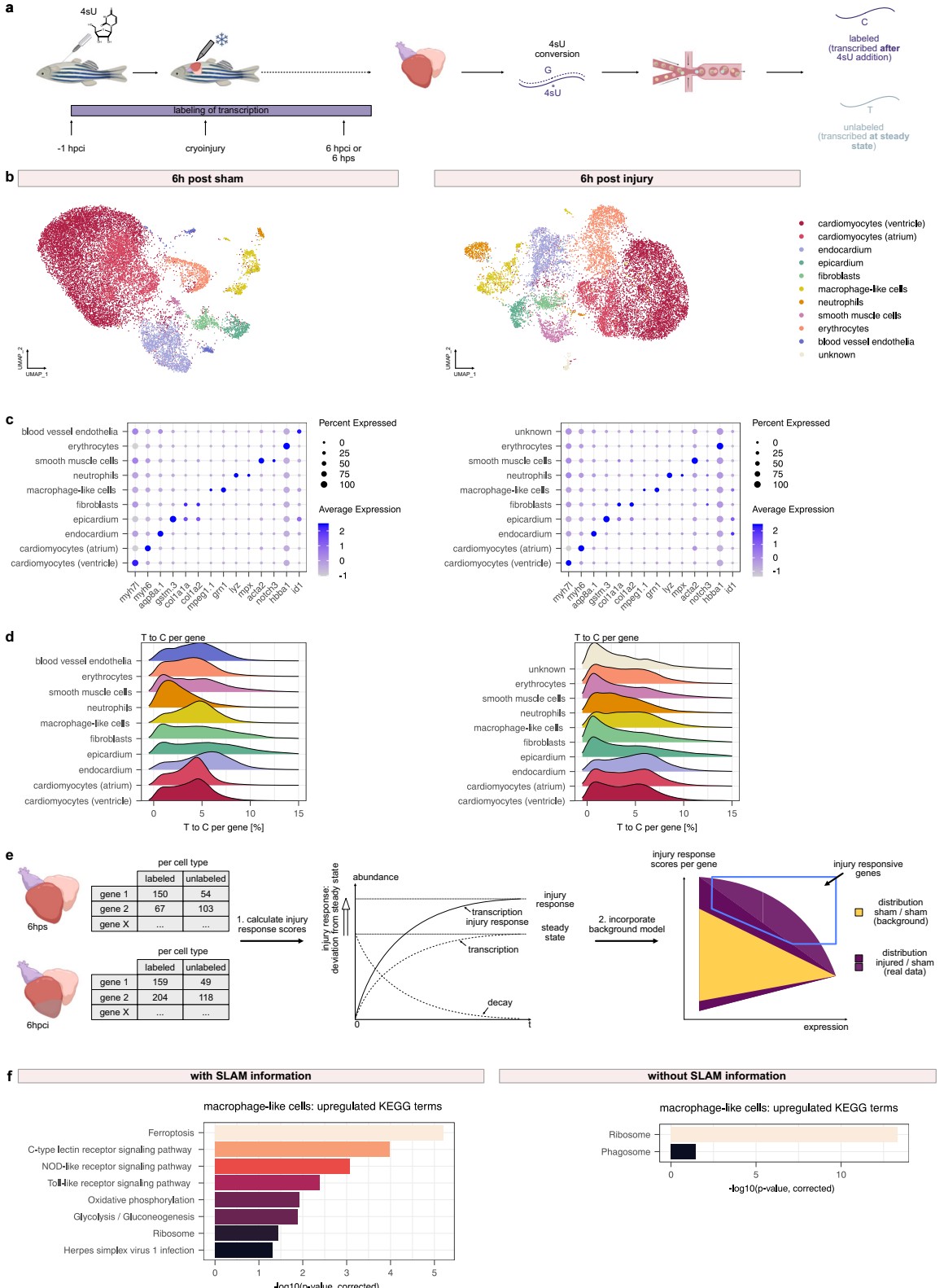

**Fig. 2 | scSLAM-seq reveals earliest transcriptional state changes after cryoinjury. a** Schematic scSLAM-seq experimental workflow. Figure created with *BioRender.com*. **b** UMAPs of 6 hps (left) and 6 hpci (right) scSLAM-seq data. 6 hps represents one 10X run of a pool of 5 biological replicates, 6 hpci represents one 10X run of a pool of 5 biological replicates. **c** DotPlots with cell type specific marker gene expression of 6 hps (left) and 6 hpci (right) scSLAM-seq data. **d** T-to-C conversion rates per gene, split by cell type in 6 hps (left) and 6 hpci (right) datasets.

**e** Schematic workflow of scSLAM-seq analysis pipeline. Figure created with *BioRender.com*. **f** Upregulated KEGG terms of injury responsive genes in macrophage-like cells using SLAM information (left) and no SLAM information, i.e., the transcriptome consisting of labeled as well as unlabeled RNA (right). Panel (**a**) was partially created in BioRender. Mintcheva, J. (2026) https://BioRender.com/5023n5v.

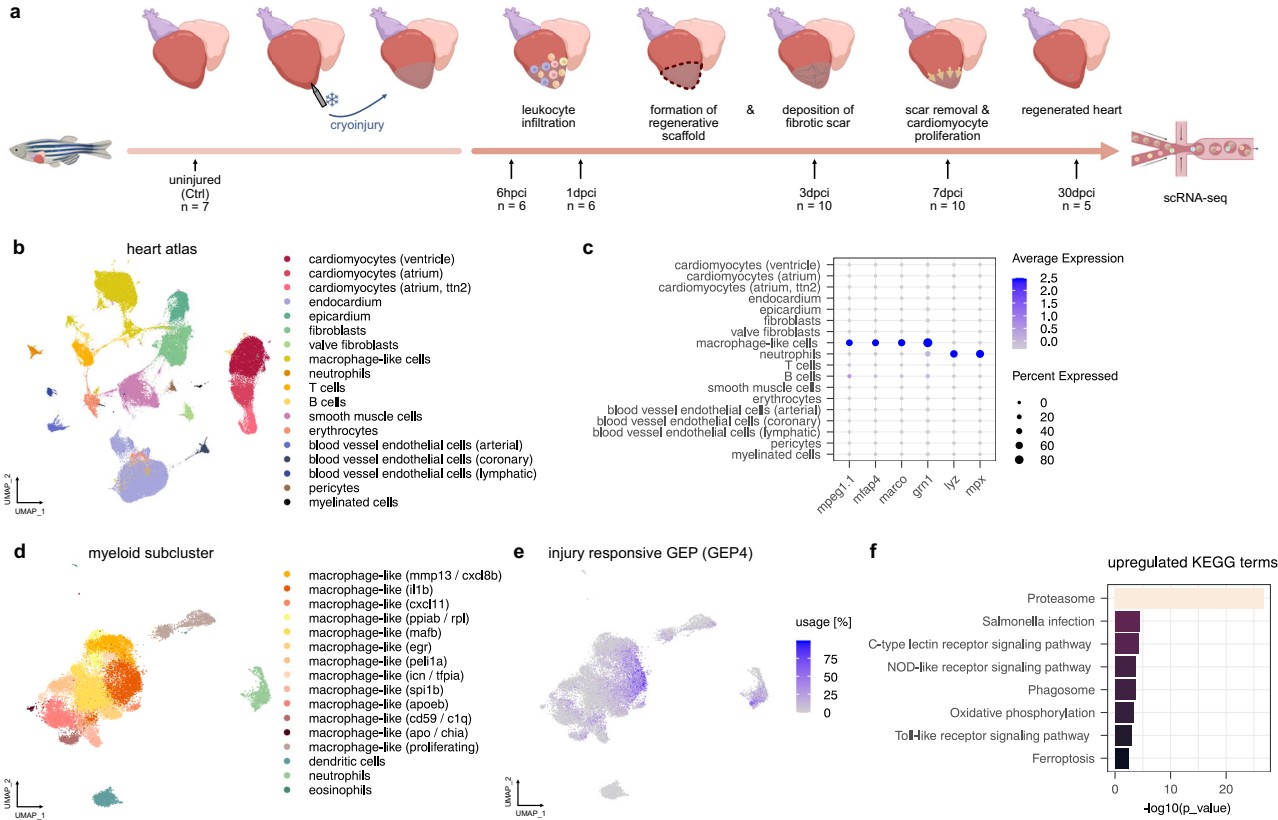

**Fig. 3 | cNMF on a scRNA-seq atlas of the regenerating zebrafish heart identifies an injury responsive gene expression program. a** Schematic representation heart regeneration hallmarks and sampling time points for scRNA-seq with their respective sample sizes. Sample sizes: *n* = 7 (uninjured), *n* = 6 (6 hpci), *n* = 6 (1 dpci), *n* = 10 (3 dpci), *n* = 10 (7dpci), *n* = 5 (30 dpci). Figure created with *BioRender.com*. **b** UMAP of the scRNA-seq heart atlas. **c** DotPlots of exemplary marker genes of macrophage-like (i.e., macrophages and monocytes) cells and neutrophils used for subclustering of myeloid cells. **d** UMAP of myeloid subcluster comprised of macrophage-like cells (macrophages and monocytes), neutrophils, dendritic cells and eosinophils. **e** cNMF usages of the injury responsive gene expression program (GEP4) in myeloid cells. **f** Upregulated KEGG terms in the injury responsive GEP. Input: genes with z-scores > mean + 2 s.d. GO term enrichment analysis was performed using g:Profiler (hypergeometric test, one-sided, g:SCS multiple testing correction). Panel (**a**) was partially created in BioRender. Mintcheva, J. (2026) https://BioRender.com/5l6uv4l.

We next investigated the half-lives of injury-responsive genes. We found that half-lives of injury-responsive genes in macrophage-like cells were only marginally shorter than those of non-injury-responsive genes. However, a comparison of the half-lives of all genes in upregulated pathways like OXPHOS, Ribosome and the TLR/NLR/CLR pathways showed markedly shorter half-lives of genes in pro-inflammatory pathways compared to Ribosome pathway genes (Supplementary Fig. 5d, e).

Of note, while our findings were largely reproducible, we also observed small differences across replicates (Supplementary Fig. 6, Supplementary Data 6). Specifically, in a replicate experiment we also observed neutrophils displaying a pro-inflammatory response by upregulating CLR and NLR pathways (Supplementary Fig. 6b), indicating either biological sample-to-sample variation or an effect of the relatively small cell numbers (e.g., only 116 and 187 neutrophils in the 6 hps datasets from Fig. 2 and Supplementary Fig. 6, respectively). In summary, our findings show that scSLAM-seq is a powerful tool to resolve short-term perturbation response, and that macrophage-like cells are one of the earliest responders to cryoinjury on the transcriptional level, by upregulating innate immune response pathways.

**The injury responsive gene expression program is characterized by TLR signaling activity in a subset of macrophage-like cells**

To further investigate and characterize the injury responsive cell state in macrophage-like cells, we made use of a large published scRNA-seq

atlas of the regenerating zebrafish heart, which we expanded by adding data for early time points like 6 hpci and 24 hpci (roughly 185,000 cells after pre-processing and filtering, including over 22,000 cells in the myeloid subcluster, Fig. 3a, b, Supplementary Data 1, 3, and 7)[8]. Specifically, we hypothesized that larger cell numbers would allow us to identify specific sub-types of injury responsive macrophages, as well as other cell types contributing to activation of damage response pathways that were not detectable in the scSLAM-seq data, e.g., because of low cell numbers. Among the myeloid cells, we detected 13 clusters of macrophage-like cells (comprised of macrophages and monocytes), including sub-types with pro-inflammatory activity, indicating substantial functional heterogeneity among macrophages (Fig. 3c, d, Supplementary Data 7). We next sought to further dissect this heterogeneity and connect it to the activation of the damage response pathways identified in Fig. 2. We therefore performed consensus non-negative matrix factorization (cNMF[40]) on the myeloid subcluster in order to identify gene expression programs (GEP) (Supplementary Fig. 7, Supplementary Data 8). In comparison to regular non-negative matrix factorization, cNMF adds a step of meta-analysis, in which multiple NMF replicates are averaged to increase robustness. Importantly, among 17 GEPs in total we identified one GEP (GEP4) which strongly resembled the pro-inflammatory injury responsive state from the scSLAM-seq analysis, consistent with our scSLAM-seq results (Fig. 3e, f). Of note, only 10% of the top genes from the injury responsive GEP overlapped with the top genes from the scSLAM-seq

(Supplementary Fig. 8). However, this low overlap is higher than the overlap with other GEPs and not unexpected due to the fundamental conceptual differences between the two approaches (e.g., cNMF also detecting sources of variation that precede cryoinjury). Importantly, GO term analysis of the overlapping genes showed that those genes were involved in the innate immune response pathways, while non-overlapping genes were involved in other processes (Supplementary Fig. 8). To further investigate the similarity of the scSLAM-seq-based injury response and the injury responsive GEP, we scored the intersecting genes in the myeloid subcluster, the entire heart atlas and the scSLAM-seq datasets (Supplementary Fig. 9). Both macrophage-like cells and neutrophils showed the highest module scores, as well as the highest upregulation at 6 hpci. A closer look at the injury responsive GEP showed that this state was predominantly used in the pro-inflammatory *macrophage-like cells (il1b)* cluster and in neutrophils, but to a lesser extent also in other clusters (Fig. 3d, e).

Activation of a GEP can, in principle, occur on two levels: either by an increase in the number of cells expressing the program, or by an increasing expression level of the program in cells that are already using the program. Here, we find a tendency for an increase in the fraction of cells using the injury responsive GEP, as well as evidence for an increase of the expression level of that GEP among macrophage-like cells that use that program at 6 hpci compared to uninjured hearts (Fig. 4a, b). To gain a more fine-grained understanding of the differential abundance of macrophage-like cell states, we performed differential abundance testing using miloR[41]. This statistical framework identifies neighborhoods in a dataset and performs differential abundance testing across conditions within those neighborhoods. Using neighborhoods rather than discrete clusters, miloR allows to more accurately pinpoint perturbed states in datasets with continuous state transitions like the activation of macrophage-like cells. With this approach, we were able to confirm that neighborhoods that are predominantly comprised of cells that use the injury responsive GEP have a positive differential abundance at 6 hpci compared to uninjured hearts (Fig. 4c). When annotating the neighborhoods by the clusters identified in Fig. 3d rather than the usage of the injury responsive GEP, we found a slight positive differential abundance in the *macrophage-like cells (il1b)* cluster and a strong positive differential abundance of neighborhoods in the *macrophage-like (mmp13/cxcl8b)* cluster (Fig. 4d). However, among the top 50 marker genes of the *macrophage-like (mmp13/cxcl8b)* cluster we were only able to detect 9 genes among the injury responsive genes in our scSLAM-seq analysis, compared to 27 out of the top 50 marker genes from the *macrophage-like cells (il1b)* cluster. This could point towards cells of the *macrophage-like (mmp13/cxcl8b)* cluster infiltrating into the heart, rather than being transcriptionally responsive to injury.

We next set out to identify transcriptional drivers of the injury responsive GEP in macrophage-like cells. We therefore plotted the cumulative expression of groups of genes. As primary candidates, we used the TLR, NLR and CLR pathways, but additionally also subdivided them into pathway layers, i.e., subsetting receptor genes as first layer, intracellular domains and first downstream signaling molecules as second layer, and molecules that are transcribed as a response to pathway activity as the effector layer (Fig. 4e, Supplementary Data 9). Among the three innate immune response pathways, we found the TLR pathway to be the most promising candidate for functional validation as proxy for the injury responsive GEP, as its expression strongly increased in the GEP at 6 hpci (Fig. 4f, Supplementary Fig. 10). When analyzing the TLR pathway in all cell types identified in the scRNA-seq heart atlas, we also found a remarkably strong response in neutrophils, and to a lesser degree in blood vessel endothelial cells and the endocardium (Fig. 4g, Supplementary Fig. 11). We speculate that we were not able to detect this response in neutrophils and blood vessel endothelial cells in the RNA labeling data because of their very low cell numbers in the scSLAM-seq dataset, which probably led to stronger

noise effects compared to cell types with several hundreds of cells. In summary, these analyses indicate that the injury responsive cell state is of a pro-inflammatory nature and that it expands in expression strength and number of macrophage-like cells immediately after cryoinjury. Moreover, TLR signaling is the most strongly upregulated pathway in the injury responsive GEP at 6 hpci and therefore a good candidate for functional validation.

Since activation of the damage response pathways is very fast, we wanted to rule out that they are partially activated by cell dissociation, a necessary step in scRNA-seq. Specifically, we were concerned that tissue dissociation may trigger an injury-like response. Furthermore, we sought to understand the spatial regulation of the pathway activations we detected with scSLAM-seq and scRNA-seq. We therefore performed hybridization chain reaction (HCR), using *mfap4.1* as a macrophage-like cell marker and *il1b* as a proxy for the injury responsive GEP, as *il1b* was one of the top upregulated genes in the GEP. We detected co-expression in HCR of 6 hpci cryosections in a substantial fraction of the *mfap4.1* positive cells (Fig. 4h, Supplementary Fig. 12a). Interestingly, the *il1b* positive *mfap4.1* macrophage-like cells did not display strong localization to the injured area at 6 hpci, which could, however, be observed at 1 and 3 dpci (Fig. 4h, Supplementary Fig. 12b). We therefore hypothesize that macrophage-like cells might localize to the injured area only later. These findings validate the existence of the pro-inflammatory, injury responsive cell state with orthogonal methods and highlight that localization of pro-inflammatory macrophage-like cells to the injured area, and macrophage-like cells in general, only occurs after 6 hpci.

## Perturbation of TLR signaling in macrophages leads to dysregulated regeneration hallmarks

We next wanted to understand the functional role of these activated pro-inflammatory macrophage-like cells, and we specifically sought to assess the contribution of the TLR signaling pathway for the early response to injury and the following regenerative cascade. Myeloid differentiation primary response 88 (MyD88) is a crucial factor in the innate immune system, downstream of TLRs[42]. MyD88 is hence an attractive target for investigating the role of TLR signaling. We first performed a re-analysis of a recently published scRNA-seq dataset of *myd88−/−* cryoinjured zebrafish hearts by Goumenaki et al.[15], in which TLR signaling is severely diminished in all cells (Fig. 5a). To interpret these data in the context of our study, we used label transfer (see "Methods") to assign cluster names from our data to the published dataset (Supplementary Fig. 13a). We focused our analysis on cells that were assigned to the *macrophage-like cells (il1b)* cluster, since activation of the injury responsive GEP was strongest in this cluster. We found that at 24 hpci, myeloid cells assigned to the *macrophage-like cells (il1b)* cluster are largely depleted in *myd88−/−* zebrafish (Fig. 5b, Supplementary Fig. 13b). Furthermore, we observed a stark decrease in the expression of TLR effectors in the *myd88* knock-out (Supplementary Fig. 13c). This observation shows that interference with the TLR pathway largely abolishes the injury responsive cell state. On the phenotypic level, Goumenaki et al. showed that lack of *myd88* led to incomplete regeneration and dysregulation of hallmarks of regeneration, and they also demonstrated that re-expression of *myd88* exclusively in the endothelial cells of *myd88−/−* zebrafish rescued some of the regeneration defects. These data further suggest that endothelial cells contribute to the early transcriptional response to heart injury.

By contrast, using another published dataset by Wei et al.[16], we found that the presence of the *macrophage-like (il1b)* cluster was expanded in macrophages after injection of clodronate liposomes, which delays macrophage recruitment (Supplementary Fig. 13d–f). This observation indicates that delaying macrophages triggers an overshoot of the pro-inflammatory response at 1 and 3 dpci, which, as reported in Wei et al., results in impaired heart regeneration. Taken together, re-analysis of these published datasets showed that global

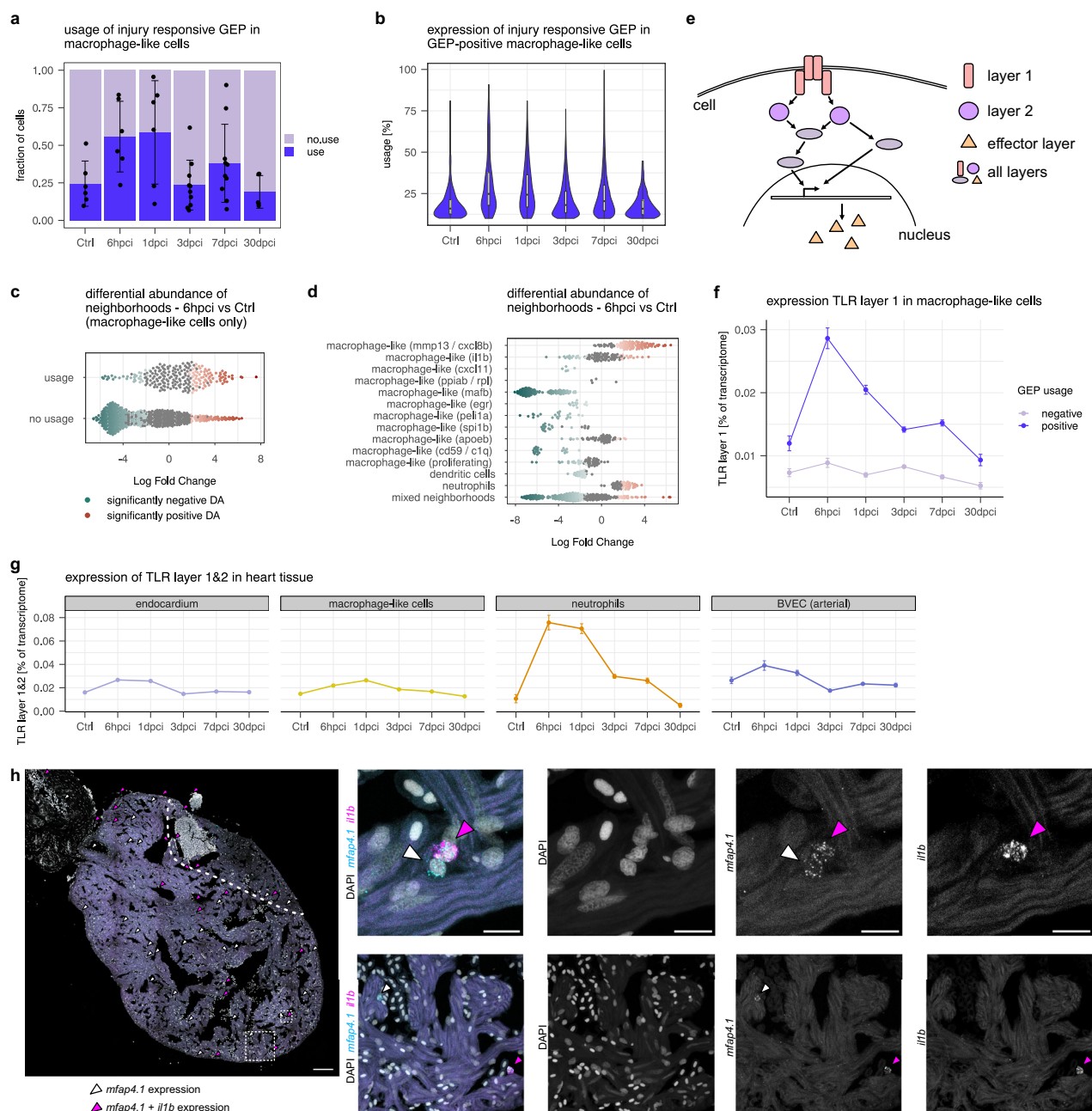

**Fig. 4 | Characterization of the injury responsive gene expression program.**
**a** Fraction of macrophage-like cells that use the injury responsive GEP (cut-off: 10% usage) by time point. Points indicate fraction of cells which use the GEP per replicate, purple bars represent average ± s.d. across replicates. Statistical test: two-sided ANOVA with Dunnett's post-hoc, comparing Ctrl to each timepoint. *P*-values: 6 hpci *p* = 0.0806, 1 dpci *p* = 0.0497, 3 dpci *p* = 1.0, 7 dpci *p* = 0.6494, 30 dpci *p* = 0.9923. **b** Usage of injury responsive GEP in macrophage-like cells which have a usage >10%. Boxes indicate the median and interquartile range (25th–75th percentiles), and whiskers extend to 1.5x the interquartile range. Statistical test: Linear mixed effects model followed by post-hoc comparisons with Ctrl against each timepoint. Two-sided *p*-values with Bonferroni multiple-testing correction. *P*-values: 6 hpci *p* = 0.0025, 1 dpci *p* = 0.505, 3 dpci *p* = 1.00, 7 dpci *p* = 0.745, 30 dpci = 1.00. **c** Differential abundance of 6 hpci versus uninjured hearts identified with miloR. Neighborhoods grouped by usage of the injury responsive GEP (cut-off: 10% usage). Red neighborhoods: significant positive differential abundance, green

neighborhoods: significant negative differential. **d** Differential abundance of 6 hpci versus uninjured hearts identified with miloR. Neighborhoods grouped by clusters of the myeloid subcluster. Red neighborhoods: significant positive differential abundance, green neighborhoods: significant negative differential abundance. **e** Schematic representation of pathway layers. **f** Average expression ± SEM of TLR layer 1 as percent of transcriptome in macrophage-like cells by time point, split by cells using or not using the injury responsive GEP (required cut-off for usage: 10%). **g** Average expression ± SEM of TLR layer 1&2 as percent of transcriptome in selected cell types of the heart atlas. **h** HCR of a 6 hpci heart with DAPI and probes against *mfap4.1* and *il1b*. Magenta arrowheads: cells that display *mfap4.1* and *il1b* expression, white arrowheads: cells that display *mfap4.1* expression only. Left: tile scan of a 6 hpci heart section, scale bar: 100 μm, dashed squares indicate areas of zoom-in for right panel, dashed line indicates injury area. Right: examples of maximum intensity projections of z-stacks of the indicated areas, scale bars: 10 μm.

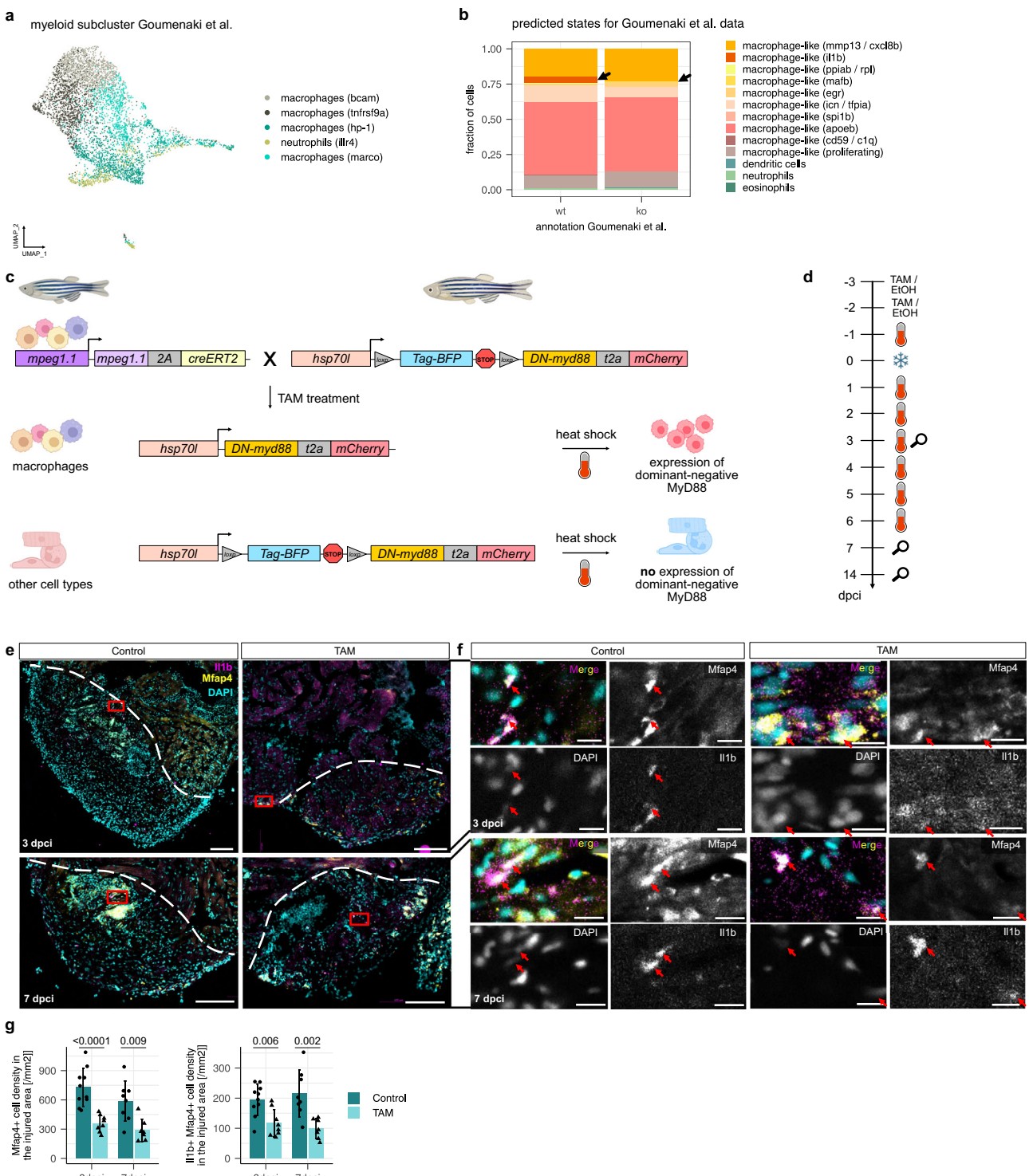

**Fig. 5 | Perturbation of TLR signaling via macrophage-specific overexpression of DN-MyD88. a** UMAP of myeloid subcluster of *myd88+/+* and *myd88-/-* hearts from Goumenaki et al. using published annotations. **b** Results of label transfer from the heart atlas myeloid data onto the data Goumenaki et al., split by wt (*myd88+/+*) and ko (*myd88-/-*) as the fraction of cells per condition assigned to clusters from our data. Black arrows point towards the fraction assigned to *macrophage-like cells (il1b)*. **c** Schematic overview of transgenic lines for macrophage-specific, inducible expression of DN-MyD88. Figure created with *BioRender.com*. **d** Experimental set-up for macrophage-specific overexpression of DN-MyD88. IP injection of TAM or EtOH as vehicle control. Thermometer indicates heat shock, snowflake cryoinjury and magnifying glass sampling timepoint. **e** Representative images of Il1b

(magenta) and Mfap4 (yellow) immunostaining of sections from cryoinjured ventricles from 3 and 7 dpci *TgKI(mpeg1:CreERT2);Tg(hsp:LSL-DN-MyD88)* zebrafish treated with vehicle or TAM. White dashed lines outline the injured area. Scale bars: 100 μm. **f** Magnified views of white-boxed region in (**e**). Arrows point to pro-inflammatory macrophages (Il1b+ Mfap4+) in the injured area. Scale bars: 10 μm. **g** Quantification of Mfap4+ cell density and Il1b+ Mfap4+ cell density in the injured area; *n* = 10 vehicle treated hearts and *n* = 8 TAM treated hearts at 3 dpci; *n* = 8 vehicle treated hearts and *n* = 7 TAM treated hearts at 7 dpci. Dots represent individual ventricles; data shown as mean ± s.d.; statistical test: two-sided Student's t-test. Panels (**c**) and (**d**) were partially created in BioRender. Mintcheva, J. (2026) https://BioRender.com/dwj9s3i.

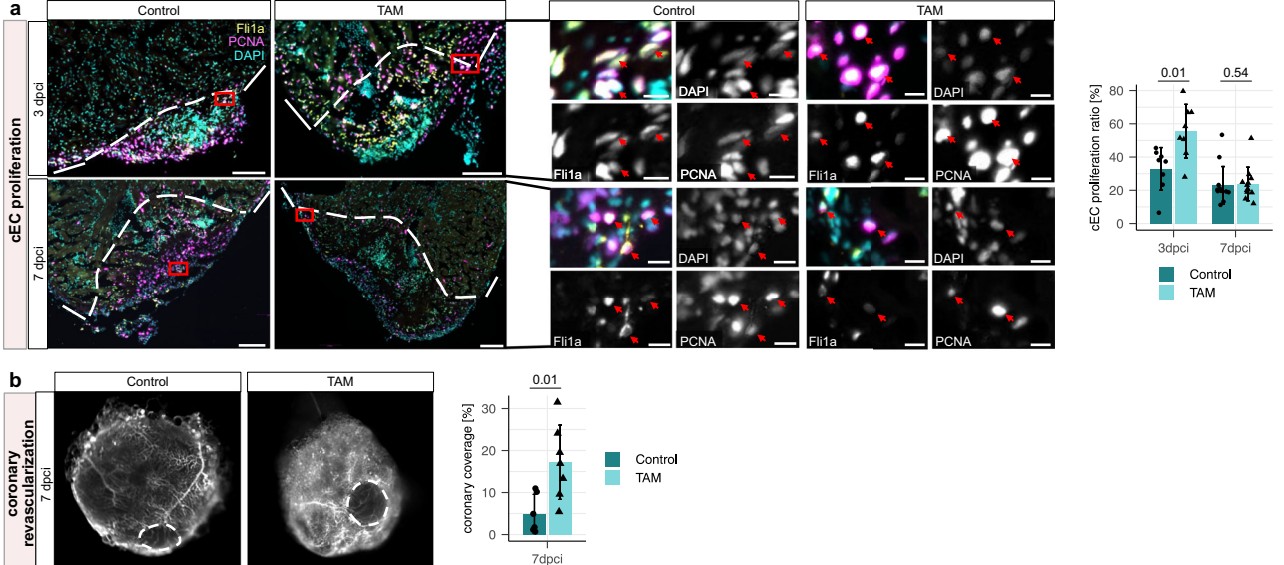

**Fig. 6 | Effect of macrophage-specific overexpression of a DN-MyD88 transgene on revascularization. a** Representative immunostaining images and quantification of cEC proliferation in the injured area of 3 and 7 dpci *TgKI(mpeg1:-CreERT2);Tg(hsp:LSL-DN-MyD88)* zebrafish treated with vehicle or TAM. White dashed lines outline the injured area. Magnified views of red-boxed regions are shown in the middle panel. Red arrows point to the proliferating cECs (Fli1α⁺ PCNA⁺), which were quantified within the epicardial region of the injured and border zone regions (indicated by white dashed lines) were quantified; *n* = 8 at 3 dpci (for both vehicle control and TAM treated); *n* = 13 vehicle treated hearts and

*n* = 12 TAM treated hearts at 7 dpci. Scale bars: 100 μm and 10 μm (magnified images). **b** Representative whole-mount images and quantification of coronary vessel coverage in 7 dpci *TgKI(mpeg1:CreERT2);Tg(hsp:LSL-DN-MyD88);Tg(−0.8flt1:RFP)* zebrafish treated with vehicle or TAM. White dashed lines outline injured area. Coverage of revascularized coronary vessels (RFP⁺) on the surface of the injured area was quantified; *n* = 6 vehicle treated hearts and *n* = 7 TAM treated hearts. The injured area is the dark space without autofluorescence at or near the apex of the heart. Scale bars: 500 μm. For all bar graphs in this figure, dots represent individual ventricles, bars represent mean ± s.d.; statistical test: two-sided Student's t-test.

reduction as well as increase of pro-inflammatory signaling have negative effects on heart regeneration, suggesting that a fine balance of pro-inflammatory signaling is crucial for successful regeneration.

Based on these observations, we hypothesized that inhibiting TLR signaling specifically in macrophages should lead to improved regenerative outcomes: While the early effects of TLR signaling (such as neutrophil recruitment) should be unaffected due to normal endothelial function, the resulting reduction of pro-inflammatory signaling at 1–3 dpci might have beneficial effects. To experimentally test this hypothesis, we generated an inducible, macrophage-specific dominant-negative MyD88 (DN-MyD88) overexpression model using the line *Tg(mpeg1.1-2A-creERT2,gcry1:NLS-EGFP)^as602*; *Tg(hsp70l:loxp-TagBFP-loxp-DN-myd88-t2A-mCherry)^bns711* zebrafish, hereafter abbreviated as *TgKI(mpeg1:CreERT2);Tg(hsp:LSL-DN-MyD88)* (Fig. 5c). This HOTcre model[43] enables heat-shock inducible expression of a DN-MyD88 exclusively in macrophages after tamoxifen (TAM) treatment (Fig. 5c). We first tested this model in larvae and were indeed able to observe successful recombination in their macrophages after TAM treatment (Supplementary Fig. 14). We then worked with adult hearts and were also able to observe successful recombination in their macrophages after TAM treatment at 3 dpci, as well as expression of the dominant-negative form of *myd88* by RT−qPCR (Supplementary Fig. 15, Supplementary Data 10). We next set out to investigate whether hallmarks of adult cardiac regeneration were affected by macrophage-specific expression of a DN-MyD88 after cryoinjury. To induce recombination, we IP injected zebrafish 3 days and 2 days prior to cryoinjury with TAM, or 25% Ethanol as a vehicle control. Starting at 1 day before cryoinjury, we heat shocked zebrafish once a day until 6 dpci, except for the day of injury to reduce the burden on the animals (Fig. 5d). This approach led to a significant reduction in both Mfap4+ cell density and Mfap4 + Il1b+ cell density in the injured area at both 3 and 7 dpci, indicating a reduction of pro-inflammatory macrophages in the injured area upon macrophage-specific DN-MyD88 overexpression (Fig. 5e–g).

To investigate the effect on regenerative processes, we first quantified coronary endothelial cell (cEC) proliferation by immunostaining for Fli1a and PCNA. We observed significantly increased cEC proliferation at 3 dpci, while the cEC proliferation ratio at 7 dpci was comparable to control samples (Fig. 6a). We also observed increased coronary vessel coverage in TAM treated zebrafish at 7 dpci (Fig. 6b). Immunostaining with Mef2 and N2.261 antibodies, which mark cardiomyocytes and de-differentiating cardiomyocytes, respectively, showed no difference in cardiomyocyte de-differentiation at 3 or 7 dpci (Fig. 7a). However, we observed a significant increase in cardiomyocyte proliferation at 7 dpci in TAM treated zebrafish by immunostaining for Mef2 and PCNA (Fig. 7b). Of note, while there were no significant differences in fibrin or collagen deposition at 7 or 30 dpci, we observed significantly reduced fibrin deposition at 14 dpci (Fig. 7c, Supplementary Fig. 16). These findings show that multiple hallmarks of zebrafish heart regeneration were improved as a result of macrophage-specific DN-MyD88 overexpression, and that some regeneration events seemed to be occurring at a slightly faster pace, given the reduced fibrin deposition at 14 dpci. However, the overall regenerative outcome (as measured by scar area in Supplementary Fig. 16d) at dpci did not seem to be significantly affected, highlighting the multifactorial nature of successful heart regeneration.

## Discussion

Here, we established in vivo single-cell RNA metabolic labeling in the adult zebrafish heart and demonstrated that scSLAM-seq provides a time-resolved readout of de novo transcription in vivo over a defined time window after injury. By separating newly transcribed from pre-existing RNA for each gene in each cell, this approach improved sensitivity for acute transcriptional activation and helped disentangle true activation events from changes in cell composition, such as recruitment or infiltration−an important distinction in highly mobile immune populations. Applying scSLAM-seq over the first 6 h after cardiac cryoinjury, we detected rapid induction of innate immune programs,

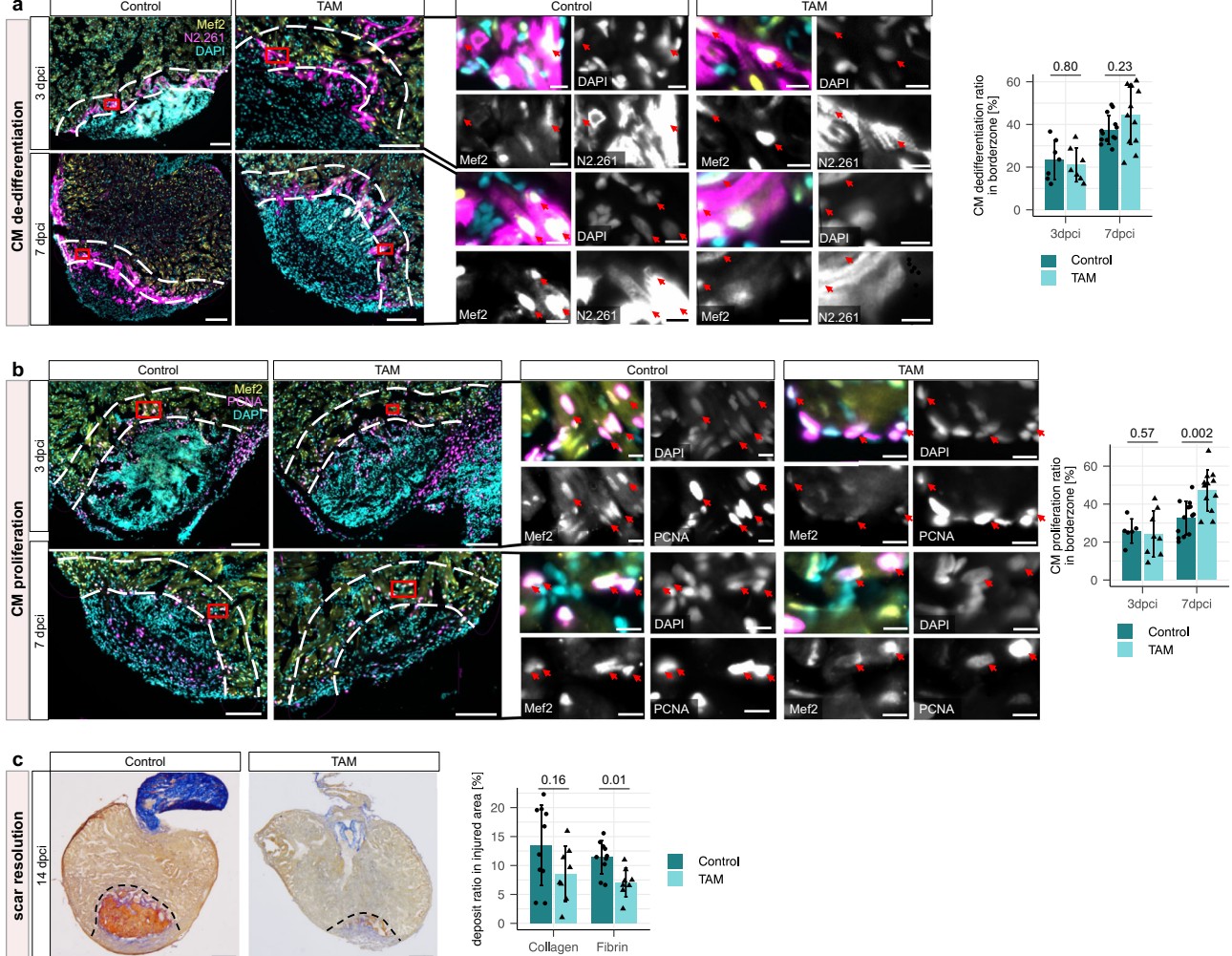

**Fig. 7 | Effect of macrophage-specific overexpression of a DN-MyD88 transgene on heart regeneration. a** Representative immunostaining images and quantification of cardiomyocyte de-differentiation in 3 and 7 dpci *TgKI(mpeg1:-CreERT2);Tg(hsp:LSL-DN-MyD88)* zebrafish treated with vehicle or TAM. White dashed lines outline the 100 μm border zone region. Magnified views of red-boxed regions are shown in the middle panel. Red arrows point to the de-differentiating cardiomyocytes (Mef2$^+$ N2.261$^+$) within the border zone which were quantified; $n = 7$ at 3 dpci (for both vehicle control and TAM treated); $n = 13$ vehicle treated hearts and $n = 12$ TAM treated hearts at 7 dpci. Scale bars: 100 μm and 10 μm (magnified images). **b** Representative immunostaining images and quantification of cardiomyocyte proliferation in 3 and 7 dpci *TgKI(mpeg1:CreERT2);Tg(hsp:LSL-DN-MyD88)* zebrafish treated with vehicle or TAM. White dashed lines outline the 100 μm border zone region. Magnified views of red-boxed regions are shown in the middle panel. Red arrows point to proliferating cardiomyocytes (Mef2$^+$ PCNA$^+$) within the border zone, which were quantified; $n = 6$ vehicle treated hearts and $n = 8$ TAM treated hearts at 3 dpci; $n = 13$ vehicle treated hearts and $n = 12$ TAM treated hearts at 7 dpci. Scale bars: 100 μm and 10 μm (magnified images). **c** Representative AFOG staining images and quantification of scar deposition in 14 dpci *TgKI(mpeg1:CreERT2);Tg(hsp:LSL-DN-MyD88)* zebrafish treated with vehicle or TAM. Black dashed lines outline the scar area. Quantification of proportions of fibrin (red) and collagen (blue) deposition in the scar area; $n = 10$ vehicle treated hearts and $n = 8$ TAM treated hearts. Scale bars: 200 μm. For all bar graphs in this figure, dots represent individual ventricles, bars represent mean ± s.d.; statistical test: two-sided Student's t-test.

including Toll-like receptor signaling and pro-inflammatory effectors, in a subset of macrophage-like cells, indicating that early myeloid transcriptional activation is a prominent component of the immediate injury response. Complementary analysis of a large scRNA-seq atlas further showed that neutrophils also mount a strong early innate response, while endothelial and epicardial populations display more modest activation, consistent with a multi-lineage early response in which macrophage-like cells (together with neutrophils) are among the earliest and most strongly activated populations at the transcriptional level.

Previous studies showed that deregulating pro-inflammatory signaling or the early macrophage response is detrimental for heart regeneration, and the overall importance of regulating pro-inflammatory signaling after cardiac damage is firmly established[10,15,16]. Guided by our time-resolved data, as well as re-

analysis of previously published datasets, we performed a macrophage-specific inhibition of the TLR adaptor MyD88, which led to the attenuation of the pro-inflammatory macrophage response and improved multiple early hallmarks of regeneration, supporting the idea that the intensity and timing of early innate immune signaling can be tuned to shape the regenerative cascade signaling[10,15]. With this concept in mind, our findings might hold translational potential for mammalian heart regeneration. In light of our data, it would be interesting to investigate if macrophage-specific fine-tuning of the pro-inflammatory response could improve regenerative outcomes in adult mice.

While we focused on the injury response in the adult zebrafish heart, our approach should be generally applicable, for instance also for increasing the signal-to-noise ratio in organoid drug screens. Our work thus exemplifies that time-resolved analysis on the single cell

level is a prerequisite for predictive models and targeted perturbations with the goal to optimize regenerative outcomes.

## Methods

### Zebrafish husbandry, handling and ethics statement
All procedures in this study adhered to the applicable ethical guidelines. Zebrafish were bred, raised and maintained following the FELASA guidelines, as well as the protocols of the Max Delbrück Center for Molecular Medicine, the Max Planck Society and the relevant local animal welfare authorities (Landesamt für Gesundheit und Soziales, Berlin, Germany, Veterinary Department of the Regional Board of Darmstadt, Darmstadt, Germany, Regierungspräsidium Karlsruhe, Germany) under the protocol numbers G0052/20, B2/2036, and 35-9185.81/G-31/23, respectively. These practices complied with the current German legislation on animal protection and the EU Directive 2010/63/EU regarding the use of animals for scientific research. Furthermore, the standards for housing and breeding were in line with the international 'Principles of Laboratory Animal Care' (NIH Publication No. 86-23, revised 1985).

### Zebrafish lines
The following wildtype and transgenic lines were used in this study: AB, *Tg(−0.8flt1:RFP)[hu5333]*, *Tg(mpeg1.1-2A-creERT2,gcry1:NLS-EGFP)[as602]* (this study), *Tg(hsp70l:loxp-TagBFP-loxp-DN-myd88-t2A-mCherry)[bns711]* (this study), *Tg(mpeg1:eGFP)[gl22]*, *Tg(mfap4:mTurquoise)[tud30244,45]* (courtesy of Nikolai Ninov's lab), *Tg(ubi:Cas9;U6:sgRNA-RFP)]/Tg(Zebrabow)[32]*.

### Generation of zebrafish transgenic lines
Sequence information was downloaded from Ensembl database (https://www.ensembl.org/index.html) (GRCz11) and further edited and analyzed with SnapGene software. To generate the *Tg(hsp70l:loxp-TagBFP-loxp-DN-myd88-t2A-mCherry)[bns711]* line, hereafter referred to as *Tg(hsp:LSL-DN-MyD88)*, the previously described *hsp70l:loxP-TagBFP-loxP-il11ra-t2a-mCherry* construct was modified[46]. The TIR domain of *myd88* coding sequence was cloned and subsequently used to replace the *il11ra* coding sequence in the construct. Cloning was performed using the In-Fusion HD Cloning Kit. 10 pg of the purified construct was delivered into zebrafish one-cell-stage WT embryos with 25 pg of *tol2* mRNA. The embryos were screened for TagBFP expression after heat-shock before being raised to the adulthood and later screened for founder identification. For the construct validation, 12.5 pg of *cre* mRNA was injected into once-cell-stage *Tg(hsp:LSL-DN-MyD88)*. *cre* mRNA injected embryos were heat-shocked and the recombination was validated by the detection of *mCherry* fluorescence.

The knock-in line *Tg(mpeg1.1-2A-creERT2,gcry1:NLS-EGFP)[as602]* (hereafter referred to as *TgKI(mpeg1:CreERT2)*) was generated using the GeneWeld CRISPR/Cas9 short homology-directed targeted integration approach as previously described to generate endogenous Cre and CreERT2 drivers[47,48]. The cassette was integrated in frame at the 3′ end of the *mpeg1.1* coding sequence using CRISPR/Cas9 and the guide RNA 5′-TTATGAGAGTAGTTGTTGATTGG-3′ (PAM underlined). 48-bp 5′ and 3′ homology arms, corresponding to genomic sequences flanking the CRISPR cut site, were first cloned into the BfuAI and BspQI restriction sites, respectively, of vector pPRISM-2A-Cre;gcry1:NLS-EGFP (Addgene#117791). To enable tamoxifen-inducible Cre activity, the *cre* coding sequence in this vector was replaced with *creERT2* from pENTR_D_CreERT2 (Addgene #27321) using NEB HiFi DNA assembly (NEB #E2621S), producing the construct pPRISM-2A-creERT2,gcry1:NLS-EGFP.

### Cardiac cryoinjuries, heat shocks, tamoxifen treatments
Zebrafish cryoinjuries were performed as described in a previously to published protocol[49]. Zebrafish were pre-sedated in Tricaine (Pharmaq Ltd) in system water. Once the zebrafish were sedated, they were placed on a sponge soaked in water from the sedation tank, ventral side facing

upwards. The heart was then exposed by making a small horizontal and vertical incision through the skin and pericardium. The ventricle was further exposed by gently pressing on the abdomen. To cryoinjure the ventricle, a liquid nitrogen precooled cryoprobe was applied to the ventricle apex. After successful cryoinjury, the zebrafish were placed in a tank with 1.5 mg/l morphine sulfate (ratiopharm) in system water. zebrafish were kept in system water supplemented with morphine for 6 h, after which they were placed back into the zebrafish facility with system water. Cryoinjuries performed in the context of the expression of the dominant-negative MyD88 overexpression were performed without morphine treatment due to different local ethical regulations.

For the macrophage-specific recombination in *TgKI(mpeg1:CreERT2);Tg(hsp:LSL-DN-MyD88)* zebrafish, adults were intraperitoneally injected with 10 µl of 1 mM Tamoxifen (Sigma H7904), or the same volume of 25% Ethanol in PBS as control at 3 days and 2 days prior to cryoinjury.

Heat shock treatment was performed by incubating the embryos or adult zebrafish in pre-warmed egg water or system water (39 °C) for 1 h. Several heat shocks were performed as described in the corresponding experimental plans.

### 4sU treatment for SLAM-seq
For SLAM-seq experiments, zebrafish were injected intraperitoneally or intrathoracically following published protocols[50,51]. In brief, zebrafish were anaesthetized with Tricaine and we injected either 25 µl intraperitoneally or 5 µl intrathoracically of a 4sU (Sigma-Aldrich), PBS, Dextran-fluorescein (Life Technologies) mix. Hearts were dissected and processed until after iodoacetamide (Sigma-Aldrich) treatment at minimal light exposure, as 4sU is a light sensitive reagent. Successful injections were verified by observation of a strong fluorescein signal.

### Bulk SLAM-seq time course
To acquire a time course with different labeling times, zebrafish were anaesthetized with Tricaine and IT injected with 5 µl of 200 mM 4sU in PBS and Dextran-fluorescein and were then placed back into a tank with fresh system water. Heart samples were acquired at 30 min, 1 h, 3 h, 6 h, 12 h, and 24 h after 4sU injection in biological triplicates and immediately homogenized in TRIzol reagent for RNA extraction and library preparation. For the 0 h timepoint we used hearts of three uninjected zebrafish.

### Bulk SLAM-seq library preparation
For bulk SLAM-seq, hearts of 4sU treated zebrafish were dissected and homogenized in tubes with ceramic beads and TRIzol reagent (Invitrogen). RNA was then extracted and libraries prepared using a modified CEL-seq protocol as previously described[34]. In brief, RNA was extracted with chloroform and precipitated with isopropanol. RNA was resuspended in RNase-free water and incorporated 4sU in the extracted RNA was converted by incubation with iodoacetamide[27]. The derivatization reaction was quenched with DTT (AppliChem) and a second RNA extraction was performed to remove excessive DTT. Next, cDNA was generated by performing reverse transcription with barcoded primers ("sample barcodes"), after which second strand synthesis was performed. cDNA was cleaned up using magnetic beads (Beckman Coulter), after which aRNA was synthesized in an in vitro transcription step. Residual primers were removed by treatment with ExoI (NEB) and rSAP (NEB). aRNA was then fragmented and cleaned up with magnetic beads. A second reverse transcription was performed on the fragmented aRNA, after which a library PCR with index primers ("index barcodes") was performed. Libraries were quantified with Qbit (Invitrogen) and the fragment distribution verified by TapeStation (Agilent Technologies). In this library layout, the transcript sequence is located on Read1 and the sample barcode and short UMI on Read2.

For our SLAM-seq time course experiment, we slightly modified the above protocol to enable UMI-based analyses. We therefore

modified primers for the first reverse transcription to additionally contain a 22 nucleotide (nt) long UMI and a 6 nt sample index. In this library layout, sample barcode and UMI were located on Read1 and the transcript sequence on Read2 (Supplementary Data 11).

Libraries were sequenced on the Illumina NextSeq500 or Nova-Seq6000 platform.

## Zebrafish heart dissociation

Zebrafish hearts were dissected and perfused in ice-cold PBS (Life Technologies) supplemented with 100 mM KCl (Carl Roth) and 60 mg/ml heparin (Sigma-Aldrich). Hearts were then dissociated following a published protocol[8]. In brief, hearts were placed in tubes with 500 μl HBSS (Gibco) supplemented with 0.001% Pluronic-F68 (Gibco) and 0.9 U/ml Liberase (Roche) and dissociated in an Eppendorf ThermoMixer F1.5 at 37 °C while shaking at 750 rpm. Dissociation was aided by intermittent pipette mixing every 7–10 min. The dissociation mix was quenched after complete dissociation, or latest after 40 min with 500 μl HBSS supplemented with 1% BSA (Sigma-Aldrich) and 0.001% Pluronic-F68. The suspension was then pelleted for 5 min at $300 \times g$ and 4 °C and washed twice with HBSS supplemented with 0.05% BSA and 0.001% Pluronic-F68. The final suspension was filtered through a 35 μm strainer (Falcon) and stained with Trypan Blue (Gibco) for live cell quantification in a counting chamber (Faust Lab Science).

## Single cell RNA-seq

ScRNA-seq was performed using the 10X Genomics 3′ Gene Expression Kit v3.1. Cell suspensions of dissociated zebrafish hearts were loaded on a 10X microfluidic chip, aiming at a recovery of 10,000 cells. Manufacturer's instructions were followed. In brief, cells were emulsified and encapsulated in GEMs together with gel beads and the reverse transcription mix. After reverse transcription and template switching, the emulsion was broken and the cDNA cleaned up. cDNA was then PCR amplified and quantified. The amplified cDNA was next fragmented, end repaired and A-tailed, followed by a bead clean-up, adapter ligation and another bead clean-up. In a last step, a sample index PCR was performed, the library bead cleaned up and quantified for sequencing. scRNA-seq libraries were sequenced on the Illumina NextSeq500 or NovaSeq6000 platforms, aiming for 20,000 reads per cell.

## Single cell SLAM-seq

To perform scSLAM-seq, we slightly modified a pre-existing protocol[32,34]. We first anaesthetized adult zebrafish and IT injected them with 5 μl of a 200 mM 4sU in PBS and Dextran-fluorescein mix and put them back into a tank with system water. One hour after injection, we anaesthetized them again to perform cryoinjury or sham injury. For sham injury, we opened the pericardium, but did not place the cryoprobe onto the ventricle. Afterwards, we placed the zebrafish in oxygenated system water supplemented with morphine for 6 h. At 6 hpci or 6 hps, we sampled the hearts, verified successful IT injection by strong fluorescein signal and proceeded to tissue dissociation as described above. Opposed to our scRNA-seq protocol, we pooled three to five hearts to compensate for cell losses during fixation and conversion steps. After tissue dissociation, we resuspended the cells in 180 μl of HBSS supplemented with 0.05% BSA and 0.001% Pluronic-F68 and added drop-wise 720 μl of ice-cold methanol while gently shaking at low intensity on the vortexer. Next, cells were fixed for 1 h at −20° °C. After fixation, 100 μl of 0.1 M iodoacetamide dissolved in 80% methanol in water was added and cells incubated at room temperature in the dark while gently rotating. The next morning, the cell suspension was cooled down on ice for 10 min, followed by a centrifugation for 10 min at $1000 \times g$ at 4 °C. The supernatant was removed, and cells gently resuspended in 500 μl quenching buffer (1 U/μl RNaseOUT [Life Technologies], 100 mM DTT, 0.1% BSA in DPBS [Gibco]). Cells were incubated for 5 min at room temperature before centrifuging for 10 min at $1000 \times g$ at 4 °C. The supernatant was removed again and cells resuspended in 60 μl of resuspension buffer (0.5 U/μl RNaseOUT, 1 mM DTT, 0.05% BSA in DPBS). Cells were counted before proceeding with scRNA-seq using the 10X Genomics 3′ Gene Expression Kit v3.1. Libraries were prepared according to manufacturer's instructions (see above) and sequenced on the Illumina NovaSeq6000 platform.

## Mapping of bulk SLAM-seq data

Bulk SLAM-seq data was mapped with STAR version 2.7.3 to the *Danio rerio* GRCz11, release 95 genome version[52].

For mapping of the bulk SLAM-seq time course, we mapped with STAR version 2.7.8a[53]. After initial demultiplexing of RPi indices with bcl2fastq, we input the sample barcodes as "cell barcodes" in order to perform the second round of demultiplexing while mapping. As the maximal possible UMI length with STARsolo is 16 nt, we used the first 16 nt as UMIs for mapping. We used a *Danio rerio* genome version GRCz11, release 92 as a mapping reference.

## Mapping and analysis of scSLAM-seq and scRNA-seq

ScSLAM-seq and scRNA-seq sequencing data was mapped with Cell-Ranger version 4.0.0, using the *Danio rerio* genome GRCz11, release 92 as a reference to which the RFP gene was added. Data was analyzed using Seurat version 4.0.6 using the standard pipeline with adapted parameters[54]. In brief, data was filtered, pre-processed, neighbors identified and cells clustered. Dimensionality reduction was performed using PCA and the UMAP algorithm and cell types identified by calculating marker genes of each cluster. For comparisons with published data, we used Seurat's label transfer algorithm to transfer our annotations to the published datasets. Marker genes were identified using Seurat's Wilcoxon rank-sum test, two-sided, with Bonferroni-adjusted p-values. Only positively enriched genes were calculated.

## Pre-processing of SLAM-seq data and extraction of SLAM information

T-to-C conversions in NGS sequencing data of SLAM-seq libraries were identified using a previously published computational pipeline using pysam version 0.16.0.1 and samtools version 1.13[34].

For bulk libraries, in brief, bam files were first MD tagged using samtools calmd to add information on mismatching and deleted reference bases. We then aggregated the substitution information in a so-called MT tag which holds the genome-wide location, substitution type and sequencing quality for each substitution. Reads were finally annotated using Rsubread.

For scSLAM-seq libraries, in brief, the bam file was first subsetted for cell barcodes in the filtered transcriptomic dataset. Next, the MD tag and global MT tag were added. Based on the global MT tag, percent T-to-C conversions could be calculated on a pseudo-bulk, per gene and per cell level. For transcript-based analyses, two separate bam files were generated containing labeled and unlabeled transcripts each. If not otherwise mentioned, we used a Q20 quality filter and at least one T-to-C conversion as a requirement to call a transcript labeled. Lastly, a count matrix was generated from each bam file.

Optionally, SNV filtering was performed by blacklisting genomic positions with frequent T-to-C conversions. As previously described, positions with at least 25% of the reads containing a T-to-C conversion compared to the reference were annotated as a SNV and blacklisted[34]. Additionally, positions with less than 10 reads coverage were disregarded as well.

## cNMF

Consensus non-negative matrix factorization was performed as previously described using version 1.3.4[40]. We performed cNMF on the non-harmonized count matrix of the myeloid subcluster, using the following parameters: n_iter=100, num_genes=3000, beta_loss=-frobenius, k = 17, d = 0.05. For visualization purposes, the usages

output was converted to percent. To classify a cell as "using" a gene expression program, we used 10% usage as a threshold. GO term analysis was performed using the R package gprofiler2, using a hypergeometric test, one-sided, g:SCS with multiple testing correction[55]. As input, we used genes with z scores larger than the fitted mean plus two standard deviations of each gene expression program.

## Differential abundance testing with miloR

Differential abundance testing was performed using the R package miloR version 1.0.0 as previously described[41].

For the myeloid subcluster, buildGraph with k = 80 and d = 20 and makeNhoods with prop=0.1 were used on the harmonized data to approximate an average neighborhood size of 200 cells, as well as a filter of 0.7 to assign mixed clusters. Only pairwise comparisons between uninjured hearts and a specific timepoint post injury were made.

To perform miloR on the injury-responsive GEP on the myeloid cells, we assigned cells either as "GEP expressing" or "not GEP expressing" with a cut-off of 10% usage. MiloR was then performed using a buildGraph with k = 80 and d = 20 and makeNhoods with prop=0.1 were used on the harmonized data to approximate an average neighborhood size of 200 cells, as well as a filter of 0.7 to assign mixed clusters. Only pairwise comparisons between uninjured hearts and a specific timepoint post injury were made.

## Hybridization chain reaction and imaging

The hybridization chain reaction (HCR) probe against *mfap4.1* was designed using the insitu_probe_generator (GitHub rwnull/insitu_probe_generator)[56]. The spliced mRNA sequences were acquired from NCBI and used as input. For probe generation, the entire mRNA sequence was used, and homopolymers up to 4 nucleotides were allowed. Probes were BLASTed against the *Danio rerio* genome GRCz11 (release 92) and low-quality probes removed. Probe sequences for *mfap4.1* were ordered as an oPool from Integrated DNA Technologies, probe sequences are reported in Supplementary Data 11. The *il1b* probe was ordered from Molecular Instruments, Inc.

HCR was performed on 10 μm cryosections of PFA (Electron Microscopy Science) fixed hearts. Sections were placed on Superfrost™ Plus Adhesion slides (epredia), dried for 30 min at 37 °C and stored at −80 °C until further use. For HCR, the manufacturer's instructions were followed. In short, sections were post-fixed in 4% PFA, followed by multiple steps of serial dehydration and rehydration. Slides were pre-hybridized for 10 min at 37 °C in hybridization buffer, followed by 24 h of probe hybridization at 37 °C in a humidified chamber using 1.6 pmol per probe in a volume of 100 μl. The next day, slides were washed with decreasing gradients of wash buffer in 5X SSCT (Sigma-Aldrich) buffer at 37 °C, followed by a short wash in 5X SSCT buffer at room temperature. The slides were then pre-amplified for 30 min at room temperature in amplification buffer, followed by 24 h of amplification with snap-cooled hairpins. In a last step, slides were washed multiple times with 5X SSCT buffer and mounted in ProLong Gold with DAPI (Invitrogen). They were sealed with nail polish and stored at 4 °C until imaging.

HCR samples were imaged on the Leica SP8 Stellaris confocal microscope with 63X magnification, and 3x Frame Average parameter. First, a tile scan of each section was acquired. Then, z stacks of areas of interest, i.e., where *mfap4.1* positive macrophages were detected in the tile scan, were acquired to obtain a more confident classification on if a cell expressed *il1b*. From each sample, 4–6 sections were imaged to obtain a more confident estimate on the fraction of *il1b* positive or negative macrophages.

## Histological analysis and imaging

For the AFOG staining and immunofluorescence staining, extracted zebrafish hearts were fixed in 4% paraformaldehyde (PFA) for 1 h at room temperature and then washed with PBS 3 times before being placed into sucrose solution (30% w/v in PBS) overnight at 4 °C. Hearts samples were then embedded in O.C.T. (Tissue-Tek) the next day and stored at −80 °C until further use. 12-μm-thick cryosections were collected on SuperFrost Plus slides (Thermo Fisher Scientific) using the Leica CM1950 cryostat and stored at −20 °C. For AFOG staining, after thawing the slides at room temperature for 15 min, the O.C.T was removed by 2 times PBST (0.2% Triton X-100) washings. The slides were further fixed in Bouin´s fixative for 2 h at 60 °C and stained following the instructions of the AFOG staining kit (BioGnost) without hematoxylin solution[2]. Stained slides were imaged under Nikon SMZ25 stereo microscope coupled with a Nikon Digital Sight DS-Ri1 camera.

For immunofluorescence staining, the slides were thawed at room temperature for 15 min and O.C.T was washed away using PBST. The slides were then boiled in sodium citrate solution (10 mM Sodium Citrate, 0.1% triton X-100, pH 6.0) at 95 °C for 10 min for antigen retrieval when using the anti-MEF2 antibody before incubation in the permeabilization solution (3% $H_2O_2$ in methanol) for 30 min at room temperature. Sections were further incubated in the blocking solution (1× PBS, 2% (v/v) goat serum, 0.2% Triton X-100 and 1% dimethyl sulfoxide) for 1 h at room temperature, and then incubated in the blocking solution with primary antibodies at 4 °C overnight. After rinsing with PBST 3 times 10 min each, the sections were further incubated with blocking solution with secondary antibodies for 3 h in room temperature. In the end, the sections were washed with PBST 3 times 10 min each and then incubated in PBST with DAPI (1:10,000 dilution, Sigma-Aldrich) for 5 min in room temperature before mounting with Fluoromount-G™ (Invitrogen) for storage at 4 °C and imaging. For immunofluorescence staining of Mfap4 and mCherry, Mfap4 antibody was used at the first round of staining. After the secondary antibody and 3 washes with PBST, sections were incubated in the blocking solution for 1 h at room temperature. Then, the primary antibody against mCherry was used for the second round of staining.

Primary antibodies used were as followed: anti-Fli1 (clone EPR4646) at 1:200 dilution (rabbit, cat. no. ab133485, Abcam); anti-PCNA (clone PC10) at 1:200 dilution (mouse, cat. no. sc-56, Santa Cruz Biotechnology); anti-MEF2 at 1:150 dilution (rabbit, cat. no. DZ01398, Boster Bio); N2.261 at 1:20 dilution (mouse, developed by H. M. Blau and obtained from the Developmental Studies Hybridoma Bank), anti-IL1 beta at 1:200 dilution (mouse, cat. no. GTX634188, Genetex); anti-Mfap4 at 1:200 dilution (rabbit, cat. no. GTX132692, Genetex); anti-GFP at 1:200 dilution (chicken, Aves Labs, GFP-1010); anti-DsRed at 1:200 dilution (recognizing mCherry, Living Colors, rabbit, Takara Bio, 632496).

Secondary antibodies used (all at 1:400 dilution) were: anti-mouse IgG (H + L) Alexa Fluor 568 (goat, cat. no. A-11004, Invitrogen); anti-chicken IgG (H + L) Alexa Fluor 488 (Invitrogen, A-11041); anti-rabbit IgG (H + L) Alexa Fluor 568 (goat, cat. no. A-11036, Invitrogen) and anti-rabbit IgG (H + L) Alexa Fluor 647 (goat, cat. no. A-21244, Invitrogen).

Imaging was performed using a Nikon Ni-E Eclipse widefield microscope equipped with a SlideExpress 2 slideloader (Märzhäuser), a SOLA Light Engine (Lumencor) and a DS-Qi2 Mono Digital Microscope Camera (Nikon). Whole-mount ventricle imaging was performed using a Nikon SMZ25 microscope coupled with a Nikon Digital Sight DS-Ri1 camera and the NIS Elements v.4.30 software.

## Quantification and statistical analysis of immunofluorescence images and AFOG stainings

The quantification of sections was done in 3–5 non-consecutive sections per ventricle and analyzed with the ZEN Blue Edition software. For CM de-differentiation and CM proliferation, the ratio of N2.261+ and PCNA + CM (MEF2 +) was calculated in the peripheral border zone areas (100 μm). For the cEC proliferation analysis, the percentage of PCNA+ cECs (Fli1α+ in the epicardium) was quantified in the injured

area and border zone. For pro-inflammatory macrophages, Mfap4+ and Mfap4 + Il1b+ cells in the injured region were counted and further normalized by the size of the injured area to obtain the cell density in the injured region. For the fibrin and collagen analysis, the AFOG images were analyzed with ImageJ (v.1.54p) to measure the injured area, fibrin area, and collagen area in all the sections with injury.

All statistical analyses were performed in GraphPad Prism (v.10.4.1). The distribution of data in each group was testing by Shapiro-Wilk normality test before the statistical analysis. The normally distributed data was analyzed with two-tailed Student's t-test, and the data failed from the normality test was analyzed with two-tailed Mann-Whitney U-test. The significance level was set to 0.05 for all the tests. The exact P values are indicated in the figures. The error bars represent the mean ± s.d.

## RT−qPCR

RNA was extracted from cryoinjured ventricles using the TRIzol Reagent (Invitrogen) with phenol-chloroform purification protocol. At least 250 ng of total RNA per sample was used for the reverse transcription with the Maxima First Strand cDNA Synthesis Kit (Thermo Fisher Scientific) according to the manufacturer's guidance. All reactions were performed with 3 technical replicates using the SYBR Green PCR Master Mix (Thermo Fisher Scientific) on the CFX Connect Real-Time System (CFX Manager 3.1, Bio-Rad Laboratories) with the following program: preamplification at 95 °C for 7 min followed by 39 cycles of amplification at 95 °C for 5 s and 60 °C for 20 s, using a melting curve from 60 to 92 °C with an increment of 1.0 °C every 5 s. Gene mRNA levels were normalized against the *rpl13a* mRNA levels and fold changes were calculated using the $2^{-\Delta\Delta Ct}$ method. The RT−qPCR primer sequences are listed in Supplementary Data 11.

## Statistics and reproducibility

Data collection and analyses were not performed blind to the conditions of the experiment. Sample sizes are indicated either in the Figure, the respective Figure legend, or in this section. Quantification of each experiment is described in the respective "Methods" or Supplemental Methods section together with the software used.

Experiments in Fig. 1b used IT injected zebrafish (50 mM $n = 1$, 100 mM $n = 2$, 200 mM plus cryoinjury $n = 2$, 200 mM plus sham injury $n = 2$) and IP injected zebrafish (200 mM $n = 3$) with $n$ being biologically independent samples from one independent experiment. Uninjected samples and 6 h 200 mM 4sU acquired in the context of the SLAM-seq time course. Experiments in Fig. 1c−e and Supplementary Fig. 1 were acquired in biologically independent triplicates per timepoint in one independent experiment. For experiments in Fig. 2, Supplementary Figs. 2a, b, d and 4a−c, we used 6 hpci ($n = 1$) and 6 hps ($n = 1$) which are both biologically and experimentally independent. For each sample, we pooled 5 hearts of biologically independent experiments. For experiments in Supplementary Fig. 5 we show independent scSLAM-seq biological replicates of 6 hpci ($n = 1$) and 6 hps ($n = 1$) which are each biologically and experimentally independent. For each sample, we pooled 5 hearts of biologically independent experiments. Experiments in Supplementary Fig. 2c used multiple organs of IP injected zebrafish (200 mM $n = 3$) with $n$ being biologically independent samples from one independent experiment. For experiments used in Fig. 3, Supplementary Figs. 6 and 7, Fig. 4a−d, g, Supplementary Figs. 9 and 10, we used published data (see Data availability) plus additional uninjured hearts ($n = 3$) of 3 independent biological samples from 3 independent experiments, 6 hpci ($n = 6$) of 6 independent biological samples of 4 independent experiments, and 1 dpci ($n = 6$) of 6 independent biological samples of 4 independent experiments. For HCR experiments in Fig. 4h, were used two biologically independent samples in two independent experiments (only one sample shown) and imaged 3 and 6 sections each. Experiments in Fig. 5e−g used 3 dpci TAM treated ($n = 8$), and vehicle control treated ($n = 10$), as well as 7

dpci vehicle control treated ($n = 8$) and TAM treated ($n = 7$) ventricles with n being biologically independent samples. Experiments in Fig. 6a used 3 dpci TAM treated ($n = 8$) and vehicle control treated ($n = 8$), as well as 7 dpci TAM treated ($n = 12$) and vehicle control treated ($n = 13$) ventricles with n being biologically independent samples. Experiments in Fig. 6b used 7 dpci TAM treated ($n = 7$) and vehicle control treated ($n = 6$) ventricles with $n$ being biologically independent samples. Experiments in Fig. 7a used 3 dpci TAM treated ($n = 7$) and vehicle control treated ($n = 7$), as well as 7 dpci TAM treated ($n = 12$) and vehicle control treated ($n = 13$) ventricles with n being biologically independent samples. Experiments in Fig. 7b used 3 dpci vehicle control treated ($n = 6$) and TAM treated ($n = 8$), as well as 7 dpci vehicle control treated ($n = 13$) and TAM treated ($n = 12$) ventricles with $n$ being biologically independent samples. Experiments in Fig. 7c used 14 dpci TAM treated ($n = 8$) and vehicle control treated ($n = 10$) ventricles with n being biologically independent samples. Experiments in Supplementary Fig. 12b−e each used $n = 4$ ventricles, with n being biologically independent samples. Experiments in Supplementary Fig. 15d used 3 dpci TAM treated ($n = 4$) and vehicle control treated ($n = 4$) ventricles with $n$ being biologically independent samples. Experiments in Supplementary Fig. 16b used 7 dpci TAM treated ($n = 9$) and vehicle control treated ($n = 7$) ventricles with $n$ being biologically independent samples. Experiments in Supplementary Fig. 16c, d used 30 dpci TAM treated ($n = 6$) and vehicle control treated ($n = 4$) ventricles with n being biologically independent samples.

## Graphics

Figure graphics were created AffinityDesigner 2. Zebrafish, cell types, injection, cryoinjury and zebrafish heart icons in Figs. 1a, 2a, 3a and 5c, d, Supplementary Figs. 15a and 16a were generated with BioRender.com and further edited, modified, color-adapted and expanded into the respective figure panel using AffinityDesigner 2. Publication licenses were therefore only acquired for the respective icons used.

## Reporting summary

Further information on research design is available in the Nature Portfolio Reporting Summary linked to this article.

## Data availability

The scRNA-seq data used in this study are available in the Gene Expression Omnibus database under accession codes GSE159032, GSE158919, GSE262247 and the NCBI SRA database Accession number PRJNA900299. Cell annotations of scRNA-seq data of Goumenaki et al. and Wei et al. were kindly provided by the authors. The scRNA-seq, bulk SLAM-seq and scSLAM-seq data generated in this study have been deposited on the Gene Expression Omnibus database under the accession code GSE305090, GSE305094, and GSE305095. Source data are provided with this paper.

## Code availability

Code used for genomics analyses as well as detailed protocols are deposited on GitHub (https://github.com/jmintch/heart_regen_slam) and Zenodo (https://doi.org/10.5281/zenodo.19256804)[57].

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

## Acknowledgements

We acknowledge MDC/BIMSB core facilities (zebrafish, light microscopy, genomics), and we thank Jeffrey J. Essner and Muhammad Naeem for contributing to the generation and characterization of the *Tg(mpeg1.1-2A-creERT2,gcry1:NLS-EGFP)^as602* line. We further thank Jana Richter for help with zebrafish experiments, Bastiaan Spanjaard for valuable discussions on the scSLAM-seq background model and Eymen Kilic for piloting miloR analyses in the scope of a lab rotation. This work was supported by the funds from the Helmholtz Association and the Max Planck Society, by the European Research Council under the European Union's research and innovation programs (ERC-CoG 101043364 to J.P.J., AdG 694455-ZMOD and AdG 101021349-TaaGC to D.Y.R.S.), *the Deutsche Forschungsgemeinschaft* (DFG grant 458913362 to J.P.J. and N.N., and DFG grant 415464617 to N.N.), the Leducq foundation, the Helmholtz Association, ERC-RA-0047 to D.P., funds from the *Deutsches Zentrum für Herz-Kreislauf Forschung* (DZHK) 81Z0100103, 81X3100110 to D.P., European Union, Grant agreement Nr. 101169349 to D.P., and a TAIWAN (NSTC) – LITHUANIA (LMT) joint research grant (NSTC 113-2923-B-001-003-MY2) to S.S.L., National Institutes of Health OD/ORIP (R24 OD036201 to M.M.). J.M. was supported by a PhD fellowship from *Studienstiftung des Deutschen Volkes*. A.S. was supported by an EMBO Long-Term Fellowship (ALTF 972-2022) and a Marie Skłodowska-Curie Actions Postdoctoral Fellowship (ID 101106181 sc-LAB2FATE) from the European Union's Horizon Europe program.

## Author contributions

Conceptualization and study design: J.P.J. and J.M.; investigation: J.M., T.L.T., P.G., A.N., A.S., R.S., S.L., Z.M., K.L.L., and A.H.; formal analysis: J.M., T.L.T, P.G., A.N., A.S., Z.M., and K.L.L.; study supervision: J.P.J., D.Y.R.S., D.P., S.S.L., M.M., A.B., and N.N.; writing—original draft: J.P.J. and J.M. in close interaction with T.L.T. and D.Y.R.S.; writing—editing and review: all authors.

## Funding

## Competing interests

The authors declare no competing interests.

## Additional information

[1]Laboratory for Quantitative Developmental Biology, Berlin Institute for Medical Systems Biology, Max Delbrück Center for Molecular Medicine in the Helmholtz Association, Berlin, Germany. [2]Humboldt University of Berlin, Berlin, Germany. [3]Department of Developmental Genetics, Max Planck Institute for Heart and Lung Research, Bad Nauheim, Germany. [4]Berlin Institute of Health at Charité, Berlin, Germany. [5]Laboratory for Electrochemical Signaling in Development and Disease, Max Delbrück Center for Molecular Medicine in the Helmholtz Association, Berlin, Germany. [6]Institute of Biomedical Sciences, Academia Sinica, Taipei, Taiwan. [7]Department of Genetics, Development and Cell Biology, Iowa State University, Ames, IA, USA. [8]Center for Regenerative Therapies TU Dresden, Dresden, Germany. [9]Paul Langerhans Institute Dresden of the Helmholtz Center Munich at the University Hospital Carl Gustav Carus of TU Dresden, German Center for Diabetes Research, Dresden, Germany. [10]Mechanisms of Cardiac Regeneration and Repair Lab, Institute of Experimental Cardiology, Heidelberg University, Heidelberg, Germany. [11]Helmholtz-Institute for Translational AngioCardioScience (HI-TAC) of the Max Delbrück Center for Molecular Medicine in the Helmholtz Association (MDC), Heidelberg/Mannheim, Germany. [12]German Center for Cardiovascular Research (DZHK), Site Heidelberg/Mannheim, Heidelberg, Germany. [13]University Hospital Schleswig-Holstein, Department of Congenital Heart Defects and Pediatric Cardiology, Kiel, Germany. [14]German Center for Cardiovascular Research (DZHK) Site Hamburg/Kiel/Lübeck, Lübeck, Germany. [15]German Center for Cardiovascular Research (DZHK) Site Rhein/Main, Frankfurt, Germany. [16]Cardio-Pulmonary Institute (CPI), Bad Nauheim, Germany. [17]German Center for Cardiovascular Research (DZHK), Site Berlin, Germany. [18]Charité – Universitätsmedizin Berlin, Berlin, Germany. ✉e-mail: janphilipp.junker@mdc-berlin.de

