## [Transparent Peer Review file · Nature Communications]

In vivo single-cell RNA metabolic labeling resolves early transcriptional responders in the regenerating zebrafish heart

Corresponding Author: Professor Jan Philipp Junker

Version 0:

Reviewer comments:

Reviewer #1

(Remarks to the Author)

This manuscript by Mintcheva et al. investigated the earliest cellular and molecular responses during adult zebrafish heart regeneration. The authors apply an innovative in vivo single-cell RNA metabolic labeling strategy (scSLAM-seq) to identify the first responder cells and their induced processes after heart injury. Their analyses suggest that a subset of macrophages rapidly activate damage response pathways, including Toll-like receptor signaling and pro-inflammatory gene expression, positioning them as sentinel cells in the regenerative process. This finding is further supported by analysis of published single-cell transcriptomic datasets. Finally, the authors focus on MyD88, a downstream mediator of Toll-like receptor signaling, and demonstrate that its inhibition enhances certain hallmarks of regeneration. The study introduces a technically powerful approach for probing acute injury responses at single-cell resolution. The topic is exciting, and the methodology is innovative. However, the work is premature and lacks sufficient depth, breadth, and validation to provide new mechanistic understanding. Specifically:

1. The sequencing data did not yield meaningful insights beyond known pathways or cell states. The claim that macrophages represent the first responders is not novel without clear mechanistic insights. Other cell types, such as endocardial and endothelial cells, also respond rapidly after injury, but were not analyzed here.
2. As the authors noted, MyD88 signaling has been previously studied in zebrafish heart regeneration, and the current work does not uncover fundamentally new mechanisms.
3. Intrathoracic injection itself is well known to activate regenerative programs. Although a sham control was included in the study, this approach may mask true early injury responses.
4. The identified *il1b+* *mfap4.1+* macrophage-like “first responder” cells are not localized to the injury site at 6 hpci, suggesting they may not be the earliest responders. Have the authors checked uninjured hearts for this population? A more thorough time course (e.g., 1 and 3 dpci, in addition to 6 hpci) is needed to evaluate the dynamics of this population.
5. Figure 5e: the images are low resolution. It is hard to recognize PCNA+ endothelial cells.
6. Figure 5i: considering the large variation in injury sizes in the cryoinjury model, analyzing only 4 samples per group is not sufficient to capture meaningful changes in scar size.

(Remarks on code availability)

Reviewer #2

(Remarks to the Author)

This is an interesting manuscript that uses an innovative method (identification of newly transcribed RNA), detailed analyses

of (single cell) sequencing data, and experimental validation to arrive at novel mechanistic insight into heart regeneration. In particular, the authors show that pro-inflammatory macrophages inhibit several hallmarks of heart regeneration. While I find most of the transcriptomics data convincing, I agree with the authors that at least some key concepts that are derived from bioinformatic methods should be validated by orthogonal methods; yet, I think the authors need to improve on these data. In addition, while the experimental validation of the hypothesis that pro-inflammatory macrophages inhibit heart regeneration is conceptually very nicely done, I do not consider the data convincing enough. In particular, I think the authors should provide more evidence that their tool works (that expression of dnMyD88 is indeed macrophage-specific and reduced the pro-inflammatory state), and they need to support the conclusion that several aspects of heart regeneration are reduced with better data. Yet, if the authors can address the specific issues detailed below, I think the paper will represent an important advance in our understanding of heart regeneration, which should be interesting to the readership of Nat. Commun.

Major:

- 1) Fig. 4a, b: Can the statement „evidence for increase in the fraction of cells using the injury responsive GEP, as well as for a transient increase of the expression level of that GEP among macrophage-like cells that use that program at 6 hpci compared to uninjured hearts” be supported by evaluation of the statistical significance of differences in GEP4 usage shown in 4a and 4b?
- 2) Fig.4f: This should be improved in several ways. 1) Higher magnification representative images should be used to more convincingly show examples of single and double positive cells. 2) More importantly, the data should be quantified, that is counts of the different cell states should be presented; this will also help to assess how reliable the data are, since it seems that very few cells were detected. 3) Noninjured or sham injured hearts should be analyzed as well to confirm that the il1b+ macrophage number is actually increasing with heart injury. 4) The suggested idea that these cells localize to the wound only at later stages should be tested as well using HCR on later stages. 5) Finally, from the presented images it's unclear how the authors know where the injured area is located. Please clarify how this was identified.
- 3) Fig. S10a-e: better data should be provided to validate the transgenes, since both are newly created in this manuscript. In particular, whether the mCherry signals in the Tam-treated double transgenic embryo represent macrophages remains unclear. At the very least, higher magnification examples including brightfield should be shown to prove that it's not autofluorescence of pigment cells that is being detected in the RFP channel. Better would be co-stainings with macrophage markers.
- 4) Fig.S10g-h: also these images are not convincing. High magnification images should be presented to prove that the presumed increase in mCherry signals is seen in cells that at least by location could be macrophages. Better would be a co-staining with macrophage markers, e.g. the mfap4.1. HCR done in Fig. 4.
- 5) Can the authors show that macrophage-specific overexpression of the dnMyD88 actually reduces the pro-inflammatory phenotype of macrophages in the heart? They use il1b expression mainly as marker for that cell state, both in the (single cell) sequencing data and the HCR in situs. qPCR on whole ventricles does not show a difference, maybe HCR does? Or can they perform a scRNASeq experiment on fish expressing the construct?
- 6) Fig. 5e, g, h: images are too small and individual channels (higher magnification views) need to be shown to allow readers to validate expression patterns and co-expression of markers. in e it's unclear how the wound area was identified (presumably by DAPI pattern, which is however not shown).
- 7) Fig. 5h: n=4 for the 7 dpi Tam treated fish is too small.
- 8) Fig.5i: n= 4 for the Tam-treated fish is too small, in particular considering the high variations observed in control treated fish.
- 9) Supplementary Data Files: use of these should be improved by adding explanations to the Excel files of what the abbreviations mean and what data they actually contain. E.g. the text refers to Supplementary Data file 4 as evidence for the statement “This response may also partially arise from ambient RNA contamination from dead cardiomyocytes, which are generally the most fragile cell type to heart dissociation”. It's entirely unclear what data this file actually contains, in particular for the “summary” sheet and how the data there support this statement...

Minor:

- 1) Fig. 1D. why do the dots have different colors? What does “log1p” mean?
- 2) Fig. S1a: why does the x-axis contain numbers smaller than 0? The labeling is somewhat ambiguous: are the red-labelled data points remaining after applying the SNV filter or are they removed?
- 3) Fig. 1E: x-Axis label missing (hours)
- 4) Fig. 2d. What is the difference between these data and the ones shown in Supp 2b? Figures legends indicate it should be the same data, but the conversion rates are quite different...

5) Please rephrase "We therefore plotted the percentage of the transcriptome that is used by the expression of groups of genes", which is confusing. The figure legend is clearer...

6) Fig. 5: p-values should only be shown to significant figures. Considering the high variations observed in the data, reporting p-value to 4 decimal places does not make sense.

(Remarks on code availability)

Reviewer #3

(Remarks to the Author)

The manuscript by Mintcheva et al. presents a comprehensive dataset of adult zebrafish hearts after cryoinjury, combining bulk SLAM-seq, in vivo scSLAM-seq, and a large scRNA-seq atlas. The central biological claim, that a subset of macrophages is among the earliest responders, engaging Toll-like receptor (TLR) signaling and a pro-inflammatory program, is supported by sequencing readouts and orthogonal validation. The macrophage-specific, inducible MyD88 manipulation is intriguing and, if fully substantiated, would provide a persuasive functional link between very early innate sensing and regenerative outcomes. (I am not a domain expert in zebrafish cardiology; my comments focus on quantitative methods, statistics, and integrative genomics.) The study has strong potential, but several core claims currently rely on pathway-level concordance rather than gene-level integration, and key kinetic and labeling assumptions need to be demonstrated quantitatively.

Major comments

- The list of mRNA half-lives from SLAM-seq data is a major result but its robustness remains unclear. Please benchmark against prior half-life compendia (e.g., ortholog/homolog comparisons where available). Connect these half-lives to downstream biology in this study. For instance, do short-lived transcripts disproportionately contribute to the early response programs observed by scSLAM-seq/scRNA-seq? Without integration, the half-life table reads as a detached resource.

- Prior reports show that 4sU at similar exposure windows can perturb transcriptional profiles in mammalian systems. Please quantify whether 4sU injection alters total-RNA expression: PCA/UMAP of uninjected vs injected controls (your Fig. 1b cohort), and differential expression of total RNA for each time point vs 0 h (using the dataset underlying Fig. 1c). This is essential to rule out 4sU-induced inflammation-like artifacts.

- Key evidence for the statement that "Macrophages are among the earliest responders" from the scSLAM-seq data currently hinges on KEGG term enrichments (e.g., Fig. 2f). Pathway significance can be influenced by power, gene set size, and background choice. Please (i) visualize the specific differentially expressed ("early-response") genes in macrophage-like cells, (ii) show their regulation across other cell types, and (iii) provide effect sizes, not only P-values. In addition consider gene-set scores (e.g., TLR/pro-inflammatory signatures) rather than single genes for these analyses.

- The claim that GEP4 "strongly resembles" the scSLAM-seq program is based on overlapping KEGG terms. That is suggestive but not a validation. Please perform a joint analysis: (i) Build a common gene signature (e.g., the same early-response gene list) and score it in both datasets; (ii) Map scSLAM-seq cells onto the scRNA-seq reference (label transfer) and compare new/old RNA components for the same genes. This would substantiate that the same genes underlie the signal in both modalities.

- The statement that ~50% of mfap4.1+ cells co-express TLR-pathway markers at 6 hpci is difficult to interpret without detection baselines. Please provide: number of sections/animals, total mfap4.1+ cells counted, per-section percentages with dispersion, background rates in mfap4.1- cells, and a statistical test. Ideally, normalize for probe efficiency (e.g., by including a housekeeping control).

Major technical/methodological comments

- Suppl. Fig. 1a suggests considerable differences between red and blue points for lowly expressed genes, which seems more than "subtle".

- Clarify whether blacklisting of SNVs occurs by read counts or UMI counts, and how you handle read-level T>C inconsistency within the same UMI (e.g., majority rule, consensus thresholds).

- The supplement implies degradation rate is inferred from the decay of the unlabeled fraction. This assumes near-complete replacement of unlabeled RNA for short-lived transcripts within your 12 h window. The fosl2 fit (Fig. 1e) appears inconsistent with full disappearance. Please (i) state the identifiability conditions explicitly, (ii) show simulations probing bias when unlabeled RNA persists, and (iii) report how often modeled plateaus deviate from the assumption.

Minor (but hopefully helpful) comments

Line 110: add a reference supporting the statement about the average mRNA half-life and single-injection sufficiency.

Clarify "conversion rate" vs "labeling rate": are these identical? Is the denominator all genomic T positions covered, or only in reads assigned as labeled? Why does ~1% qualify as "low overall labeling rate" (L111)—relative to what benchmark?

Lines 169 & 172: the core kinetic calculation resembles early SLAM-seq formulations (e.g., fold-change of labeled RNA; see Muhar et al., Science 2018). The novel contribution appears to be your noise-dominated gene filtering. Consider moving the filtering rationale/results into the main text and trimming the generic derivation, or at least reframing to highlight what is new here.

Line 249: Please specify how cluster names were derived (marker-based, label transfer, manual curation).

Explain "TLR layer 1" in Figure 4e; The figure is difficult to interpret without cross-referencing Suppl. Fig. 7. Please add a schematic or brief explanation in the main figure legend about what "layer 1" denotes and how it was derived.

I suggest to avoid the term "simple back-of-the-envelope calculation" (L169);

(Remarks on code availability)

Version 1:

Reviewer comments:

Reviewer #1

(Remarks to the Author)

The authors have adequately addressed my major concerns, and I have no further comments.

(Remarks on code availability)

The provided link does not work. There is no such record on GitHub.

Reviewer #2

(Remarks to the Author)

Overall, the authors have addressed all issues raised by me in a satisfactory manner. Yet, one major problem remains: they now provide higher magnification views of their immunofluorescence experiments, which allow for assessment of the validity of their conclusions. Yet, several of these raise important questions that cast doubt on the rigor with which these stainings have been analyzed. Specifically,

- 1) Fig. 6a: why does PCNA staining in several instances appear as large "blobs" that are much larger than DAPI+ nuclei (most evident in the lower right panels).
- 2) Fig. 6a: why do arrows, which are supposed to point to PCNA+ flia+ cells, in many cases clearly NOT point to such double positive cells? This is for example very evident in the upper left (control) panel.
- 3) Fig. 6c: also here, the arrows which are supposed to point out Mef2+ N2.226+ cells/nuclei partly point to very spurious examples (e.g. upper right "TAM", right-most cell).
- 4) Fig. 6c: also here some PCNA+ nuclei are pointed out, where no DAPI staining seems to exist, e.g. lower left panel
- 5) Image quality throughout is an issue. Most channels are way too overexposed (everything except DAPI).

Rigorous quality control would 1) not overexpose channels to an extent where all subtleties of signal/noise assessment are lost 2) would exclude PCNA stains that do not overlap with DAPI from analysis, and 3) obviously would not count cells as double positive if they are not...

Overall, these issues unfortunately leave me with the impression that the paper is very strong on the transcriptomics / bioinformatics side, but less so on the experimental validation.

I cannot recommend publication until authors can show more convincing evidence of rigorous analysis of immunofluorescence data.

(Remarks on code availability)

Reviewer #3

(Remarks to the Author)

I would like to congratulate the authors on a thorough and comprehensive revision. However, one significant concern remains regarding the appropriateness of the kinetic model: The new analyses clearly show that labeled RNA plateaus at ~ 0.8 instead of 1 (which is an inherent assumption of their kinetic model). Consequently, decay rates are systematically underestimated. Based on theoretical considerations regarding the difference of expectations of the decay rates, the authors postulate that this underestimation is "moderate". However, the implications of such bias remain unclear. Specifically, while the ranking of decay rates seems to be unaffected in expectation, how does the misspecification behave in the presence of noise? To what extent are R^2 values inflated which are used for filtering, potentially biasing results based on decay rates?

I do not want the authors to provide a more in-depth theoretical analysis. Rather, it should be straight-forward to estimate the labeling correction parameter directly from the data and include it into the model. Presenting model fits that utterly deviate from data (as in Figure 1e) detracts from the high quality of this otherwise very strong manuscript, and I strongly recommend incorporating this correction.

(Remarks on code availability)

Version 2:

Reviewer comments:

Reviewer #2

(Remarks to the Author)

Authors have improved image quality and I trust that the issues with wrongly identified double-positive cells were honest mistakes and not signs of a lack of rigor that would effect the validity of the results. I can now recommend publication.

(Remarks on code availability)

REVIEWER COMMENTS

Reviewer #1 (Remarks to the Author):

This manuscript by Mintcheva et al. investigated the earliest cellular and molecular responses during adult zebrafish heart regeneration. The authors apply an innovative *in vivo* single-cell RNA metabolic labeling strategy (scSLAM-seq) to identify the first responder cells and their induced processes after heart injury. Their analyses suggest that a subset of macrophages rapidly activate damage response pathways, including Toll-like receptor signaling and pro-inflammatory gene expression, positioning them as sentinel cells in the regenerative process. This finding is further supported by analysis of published single-cell transcriptomic datasets. Finally, the authors focus on MyD88, a downstream mediator of Toll-like receptor signaling, and demonstrate that its inhibition enhances certain hallmarks of regeneration. The study introduces a technically powerful approach for probing acute injury responses at single-cell resolution. The topic is exciting, and the methodology is innovative. However, the work is premature and lacks sufficient depth, breadth, and validation to provide new mechanistic understanding.

We thank the reviewer for the careful and candid evaluation of our work, and for highlighting where additional depth and validation were needed. In response, we performed substantial additional experiments and analyses and revised the manuscript to sharpen the key advance of the study: a time-resolved, single-cell readout of *de novo* transcription during the earliest hours after injury. We strengthened biological validation by expanding orthogonal imaging (including uninjured and later time points) and by improving quantification and presentation of microscopy data. In parallel, we added analyses to address potential confounders of the experimental design (4sU/IT injection) and broadened our cross-cell-type assessment of early innate immune pathway activity in the scRNA-seq atlas, thereby placing the macrophage response in a clearer multi-lineage context.

Specifically:

1. The sequencing data did not yield meaningful insights beyond known pathways or cell states. The claim that macrophages represent the first responders is not novel without clear mechanistic insights. Other cell types, such as endocardial and endothelial cells, also respond rapidly after injury, but were not analyzed here.

We thank the reviewer for this important comment. We agree that several components of the early injury response (e.g., activation of innate immune pathways) are consistent with prior knowledge, and that multiple cardiac lineages (including endothelial/endocardial-related populations) respond rapidly after injury. The key advance of our study is that **in vivo scSLAM-seq enables a time-resolved, single-cell readout of *de novo* transcription** within a defined window (here the first 6 hours post injury), thereby separating **newly induced transcriptional activation** from (i) pre-existing transcripts and (ii) changes in cell abundance due to recruitment/infiltration.

Using this approach, we identify a rapid pro-inflammatory transcriptional activation program in a subset of macrophage-like cells within 6 hpci, characterized by upregulation of innate immune pathways including TLR signaling. Importantly, this injury-specific signal is only robustly detectable when leveraging the labeling/new-RNA information – and is not recovered to the same extent when analyzing the same data using total transcriptome counts without SLAM information (Fig. 2f right panel, Supplementary Table 5) – highlighting the benefit of the time-resolved new-transcription readout.

To address the reviewer’s concern that other early responding cell types were not analyzed, we now explicitly quantify and compare early pathway activation across cell types using our expanded scRNA-seq atlas (including 6 hpci). Specifically, when scoring TLR pathway activity across **all cell types** in the atlas, we observe that – besides macrophage-like cells – neutrophils show a remarkably strong early response, while vascular endothelial cells and the endocardium show a detectable but weaker increase at 6 hpci (Fig. R1- 1, Fig. R1- 2). We further performed gene-signature/module scoring using a shared early-response signature (overlap between scSLAM-seq injury-responsive genes and the injury-responsive gene expression program from non-negative matrix factorization) across the entire atlas, which again shows the highest induction at 6 hpci in macrophage-like cells and neutrophils (Fig. R1- 3). Together, these analyses clarify that macrophage-like cells are **not the only** responding population, but they are **among the earliest and most strongly transcriptionally activated populations** in our time-resolved (*new RNA*) readout. This information is now included in Fig. 4g and is highlighted in the Discussion of the revised manuscript.

Fig. R1- 1 Expression of TLR layer 1&2 as percent of transcriptome in selected cell types of the heart atlas. Depicted are average expression \pm SEM. These data show that myeloid cells are not the only responding population, but they are among the earliest and most strongly transcriptionally activated populations. **Now included as Fig. 4g.**

Fig. R1- 2 Expression of TLR pathway layers across all cell types of the heart scRNA-seq atlas by time point. Depicted is the average expression \pm SEM. These data show that myeloid cells are not the only responding population, but they are among the earliest and most strongly transcriptionally activated populations. In the manuscript: Supplementary Fig. 11.

Fig. R1- 3 a, Seurat module scores of the intersect of genes from injury-responsive genes identified in macrophage-like cells with scSLAM-seq and the injury-responsive GEP in the myeloid subcluster of the heart atlas. Clusters were grouped into cell types (macrophage-like cells, neutrophils, dendritic cells, eosinophils). b, Seurat module scores of the intersect of genes in all cell types of the heart atlas. c, Seurat module score of the intersect of genes in the total RNA (i.e. labeled and unlabeled) of the 6 hpci scSLAM-seq sample. d, Seurat module score of the intersect of genes in the labeled RNA of the 6 hpci scSLAM-seq sample. These data show that the intersecting genes identified by cNMF and the scSLAM-seq injury response show an upregulation both in macrophage-like cells and neutrophils. **Now included as Supplementary Fig. 9.**

And, regarding “meaningful insights beyond known pathways”: while the identity of some pathways (e.g. innate immune sensing) is expected, our work contributes (i) **direct kinetic evidence** for very early *de novo* transcriptional activation in vivo at single-cell resolution, (ii) identification and characterization of the specific **injury-responsive macrophage-like cell state** (including distinguishing transcriptional activation from abundance changes/infiltration in the broader myeloid compartment), and (iii) a **functional, cell-type-specific perturbation** motivated by these data (macrophage-restricted DN-MyD88) that alters key regeneration hallmarks. We also wish to note that we here present the first application of scSLAM-seq in a regenerative setting. We have revised the manuscript text (especially title, abstract, discussion) accordingly to emphasize this “time-resolved transcriptional activation” framework and to avoid overstating priority, using “among the earliest transcriptional responders” rather than implying that macrophages are uniquely the first responders.

2. As the authors noted, MyD88 signaling has been previously studied in zebrafish heart regeneration, and the current work does not uncover fundamentally new mechanisms.

As the reviewer correctly notes, MyD88 signaling has been studied before in zebrafish heart regeneration. Specifically, it was shown that heart regeneration is dampened in the *myd88*-/- knockout (Goumenaki et al., Nat Cardiovasc Res, 2024). However, we here show that the finetuning of MyD88 signaling, in a cell type dependent manner, is important for multiple early regenerative hallmarks. This is a fundamentally new and surprising finding: Different studies have shown that deregulating pro-inflammatory signaling and/or the early macrophage response is detrimental for heart regeneration (PMIDs: 28632131, 37498060, 39271818, 40554763), but we here show for the first time, and in a data driven approach, that it is possible to optimize early hallmarks of heart regeneration in zebrafish by manipulating MyD88 signaling. In the revised manuscript we discuss more clearly how our manuscript goes beyond these previous publications.

3. Intrathoracic injection itself is well known to activate regenerative programs. Although a sham control was included in the study, this approach may mask true early injury responses.

The reviewer raises an important point. As the reviewer correctly notes, the sham control is crucial, but there is the risk that any responses that are shared between IT injection and cardiac cryoinjury response might be masked. In general, we would expect that the magnitude of the response to cryoinjury is larger than the response to IT injection, even if the same pathways are triggered. However, we now discuss this explicitly in the manuscript as a limitation of the approach.

To further address the reviewer's comment, we now include a PCA analysis of the bulk SLAM-seq time course and differential gene expression analysis of uninjected zebrafish hearts compared to hearts labeled for 6 h with 4sU (Fig. R1- 4). We do observe an activation of known immune-related genes, however, PCA analysis together with sham and cryoinjured IT injected hearts showed that the effect of sham and cryoinjury leads to a much larger transcriptional effect than just of the 4sU IT injection, confirming the difference in effect size discussed above.

Fig. R1- 4 **d**, PCA of bulk SLAM-seq time course data based on total, i.e. labeled + unlabeled counts. **e**, Volcano plot of differential gene expression of 6 h vs. 0 h (uninjected) time course datapoints. **f**, PCA of uninjected, IT-injected 6 h, IT injected 6 hps (sham), IT-injected 6 hpci (cryoinjury) and IP injected 6 h samples based on total, i.e. labeled and unlabeled, reads. **Now included in Supplementary Fig. 1d-f.**

Furthermore, we also identify the response of e.g. TLR signaling in the larger conventional scRNA-seq dataset (Fig. 3, 4), which does not involve IT injection, thereby validating the biological finding.

4. The identified *il1b+* *mfap4.1+* macrophage-like “first responder” cells are not localized to the injury site at 6 hpci, suggesting they may not be the earliest responders. Have the authors checked uninjured hearts for this population? A more thorough time course (e.g., 1 and 3 dpci, in addition to 6 hpci) is needed to evaluate the dynamics of this population.

The main aim of our HCR experiments was to rule out that the injury-responsive cell state is a dissociation artefact, and it was not our primary goal to duplicate the full scRNA-seq time course analysis by HCR. We now discuss more explicitly in the revised manuscript.

However, we agree with the reviewer that the localization dynamics of the *il1b+* macrophages warrants further attention. We now validated our HCR experiments with immunofluorescence (IF using an anti-Il1b antibody in *mfap4:mTurquoise* reporter zebrafish), and we also added the suggested further timepoints (1 and 3 dpci, Fig. R1- 5):

In the healthy uninjured heart, we only detect very few *Il1b+* *Mfap4+* macrophages. At 6 hpci we find a few *Il1b+* *Mfap4+* macrophages in the injury area. However, there is no evident preferential localization to the injury area compared to the uninjured area, confirming the results of our HCR experiment. At 1 and 3 dpci we observe strong *Mfap4+* and *Il1b+* signals in the injured and epicardial area, indicating increasing abundance of pro-inflammatory macrophages into the injured area after 6 hpci. In summary, we observe an interesting two-step process: early delocalized activation of *Il1b* expression at 6 hpci, followed by stronger localized expression at later stages. Understanding the mechanisms that lead to the later localization (and further amplification) of *Il1b+* expression after the initial delocalized transcriptional response will be an important subject for further research.

Fig. R1- 5 Representative immunostaining images of pro-inflammatory macrophages ($Il1b^+ Mfap4^+$) in heart section from uninjured $Tg(mfap4:mTurquoise)^{tud302}$ zebrafish: uninjured, 6 hpci, 1 dpci, 3 dpci, 7 dpci with; $n = 4$ each. White dashed lines outline the injured region. Magnified views of red-boxed region are shown at right panel. White arrows point to the pro-inflammatory macrophages ($Il1b^+ Mfap4^+$). Magnified views of red-boxed region are shown at right panel. Scale bar: $100 \mu m$ and $10 \mu m$ (magnified images). **Now included as Supplementary Fig. 12b.**

5. Figure 5e: the images are low resolution. It is hard to recognize PCNA+ endothelial cells.

Image resolution was lowered in the initial submission to stay within the journal's required limits of data size. We are now providing high resolution images. Moreover, we have now increased the display size of all images of immunofluorescent stainings and included zoom-in panels for better visibility of molecule co-expression. We hope this addresses the reviewer's comment (Fig. R1- 6).

Fig. R1- 6 Layout of images formerly shown in Fig. 5. Now displayed larger and with included zoom-in panels. **Now included as Fig. 6.**

6. Figure 5i: considering the large variation in injury sizes in the cryoinjury model, analyzing only 4 samples per group is not sufficient to capture meaningful changes in scar size.

We have now increased the sample sizes for the following experiments: AFOG staining at 14 dpici (control n=10 and TAM n=8). We have further increased samples sizes for coronary endothelial cell proliferation at 7 dpici (control n=13, TAM n=12), cardiomyocyte proliferation at 7 dpici (control n=13, TAM n=12) and cardiomyocyte dedifferentiation at 7 dpici (control n=13, TAM n=12), now displayed in **Fig. 6** (see Fig. R1- 6). With these, we obtained similar results, supporting our previous conclusions, and further strengthening our observation.

Reviewer #2 (Remarks to the Author):

This is an interesting manuscript that uses an innovative method (identification of newly transcribed RNA), detailed analyses of (single cell) sequencing data, and experimental validation to arrive at novel mechanistic insight into heart regeneration. In particular, the authors show that pro-inflammatory macrophages inhibit several hallmarks of heart regeneration. While I find most of the transcriptomics data convincing, I agree with the authors that at least some key concepts that are derived from bioinformatic methods should be validated by orthogonal methods; yet, I think the authors need to improve on these data. In addition, while the experimental validation of the hypothesis that pro-inflammatory macrophages inhibit heart regeneration is conceptually very nicely done, I do not consider the data convincing enough. In particular, I think the authors should provide more evidence that their tool works (that expression of dnMyD88 is indeed macrophage-specific and reduced the pro-inflammatory state), and they need to support the conclusion that several aspects of heart regeneration are reduced with better data. Yet, if the authors can address the specific issues detailed below, I think the paper will represent an important advance in our understanding of heart regeneration, which should be interesting to the readership of Nat. Commun.

We thank the reviewer for the constructive and encouraging comments and for the clear guidance on where the study required stronger orthogonal validation and more convincing functional evidence. To address these points, we (i) added statistical testing and clearer reporting for key atlas-derived claims, (ii) substantially improved the in situ/IF validations by adding higher-magnification examples, quantification, and additional controls/time points, and (iii) strengthened the validation of our inducible macrophage-specific DN-MyD88 tool through additional imaging and adult-heart co-stainings. We also increased sample sizes for the regeneration hallmark readouts and improved figure presentation (larger images, zoom-ins, and clearer identification of relevant regions), and we added clarifying documentation for the Supplementary Data files to improve usability.

Major:

1) Fig. 4a, b: Can the statement „evidence for increase in the fraction of cells using the injury responsive GEP, as well as for a transient increase of the expression level of that GEP among macrophage-like cells that use that program at 6 hpci compared to uninjured hearts” be supported by evaluation of the statistical significance of differences in GEP4 usage shown in 4a and 4b?

We thank the reviewer for this suggestion. For Fig. 4a we first did the Shapiro-Wilk normality test to confirm that the data is normally distributed. We then performed an ANOVA test with Dunnett's multi-comparison procedure, to specifically test each time point versus the single control sample while controlling the family-wise error rate. The Dunnett-corrected p value for Ctrl vs 6hpci is $p = 0.0806$, (Ctrl vs 1 dpci: $p = 0.0499$). We now included error bars (standard deviation) and report the p values in the figure legend (Fig. R2- 1). Moreover, we clarify more clearly which effects are statistically significant in the manuscript.

Fig. R2- 1 Fraction of macrophage-like cells that use the injury responsive GEP (required cut-off: 10 % usage) by time point. **Now updated as Fig. 4a.**

For Fig. 4b (expression of injury responsive GEP in GEP-positive macrophage-like cells), we fitted a mixed effects model (to account for cell-level variability and sample-level grouping). We tested Ctrl against each time point and performed Bonferroni correction. We now report the p-values in the figure legend.

- Ctrl vs 6hpci: $p_{\text{adjusted}} = 0.002$
- Ctrl vs 1dpci: $p_{\text{adjusted}} = 0.505$

Of note, while the overall distributions for 6hpci and 1dpci in Fig. 4b look very similar, there are clear differences when plotting the average expression per sample (Fig. R2- 2), which explain the difference in p values between the two comparisons.

Fig. R2- 2 Average usage of macrophage-like cells that use the injury responsive GEP split per sample highlights higher sample variability at 1 dpci compared to 6 hpci.

2) Fig.4f: This should be improved in several ways. 1) Higher magnification representative images should be used to more convincingly show examples of single and double positive cells. 2) More importantly, the data should be quantified, that is counts of the different cell states should be presented; this will also help to assess how reliable the data are, since it seems that very few cells were detected. 3) Noninjured or sham injured hearts should be analyzed as well to confirm that the il1b+ macrophage number is actually increasing with heart injury. 4) The suggested idea that these cells localize to the wound only at later stages should be tested as well using HCR on later stages. 5) Finally, from the presented images it's unclear how the authors know where the injured area is located. Please clarify how this was identified.

1)

We now provide higher magnification of the zoom-ins, to more clearly highlights single- and double-positive signal (Fig. R2- 3).

Fig. R2- 3 HCR of a 6 hpci heart with DAPI and probes against *mfap4.1* and *il1b*. Magenta arrow heads point to cells that show *mfap4.1* and *il1b* expression, white arrow heads point to cells that show *mfap4.1* expression only. Left: tile scan of a 6 hpci heart section, scale bar: 100 μ m, dashed squares indicate areas of zoom-in for right panel, dashed line indicates injury area. Right: two examples of maximum intensity projections of z-stacks of the indicated areas in the tile scan, scale bar: 10 μ m. **Updated figure with larger zoom-in panels now included as Fig. 4h.**

2)

We have now added a quantification of *mfap4.1*+ and *mfap4.1*+ *il1b*+ cells (Fig. R2- 4). We performed HCR on n=2 6 hpci hearts and imaged 3 and 6 sections, respectively, which are now added to the Figure legend of the quantification.

Fig. R2- 4 Quantification of *mfap4.1*+ (gray) and *mfap4.1*+ *il1b*+ (pink) positive cells from HCR on 6 hpci heart cryosections. Number of sections imaged in 6 hpci #1: 6, 6 hpci #2: 3. **Now included as Supplementary Fig. 12a**

3&4)

The main aim of our HCR experiments was to rule out that the injury-responsive cell state is a dissociation artefact, and it was not our primary goal to duplicate the full scRNA-seq time course analysis by HCR. We now discuss more explicitly in the revised manuscript.

We agree with the reviewer that the localization dynamics of the *il1b*+ macrophages warrants further attention. We now validated our HCR experiments with immunofluorescence (IF using an anti-Il1b antibody in *mfap4*:mTurquoise reporter zebrafish), and we also added further later timepoints (1 and 3 dpci) to test for the localization of pro-inflammatory macrophages. In the healthy uninjured heart we only detect very few Il1b+ Mfap4+ macrophages (Fig. R2- 5). At 6 hpci we find Il1b+ Mfap4+ macrophages in the injury area, confirming the increase of the pro-inflammatory macrophage cell state at 6 hpci. However, there is no evident preferential localization to the injury area compared to the uninjured area, confirming the results of our HCR experiment. At 1 and 3 dpci we observe strong Mfap4+ and Il1b+ signals in the injured and epicardial area, indicating a strong increase in the abundance of pro-inflammatory macrophages into the injured area after 6 hpci. In summary, we observe an interesting two-step process: early delocalized activation of *il1b* expression at 6 hpci, followed by stronger localized expression at later stages. Understanding the mechanisms that lead to the later localization (and further amplification) of Il1b+ expression after the initial delocalized transcriptional response will be an important subject for further research.

Fig. R2- 5 Representative immunostaining images of pro-inflammatory macrophages ($Il1b^+ Mfap4^+$) in heart section from uninjured $Tg(mfap4:mTurquoise)^{tud302}$ zebrafish: uninjured, 6 hpci, 1 dpci, 3 dpci, 7 dpci with; $n = 4$ each. White dashed lines outline the injured region. Magnified views of red-boxed region are shown at right panel. White arrows point to the pro-inflammatory macrophages ($Il1b^+ Mfap4^+$). Magnified views of red-boxed region are shown at right panel. Scale bar: $100 \mu m$ and $10 \mu m$ (magnified images). **Now included as Supplementary Fig. 12b.**

5)

We agree with the reviewer that the location of the injury area is not obvious in the microscopy image. To address this issue, we performed spatial transcriptomics (Open-ST method; Schott et al., Cell, 2024) of an adjacent section. The plots below (Fig. R2- 6) show that the injury area can be identified clearly in the Open-ST data based on the following criteria: reduced *myl7* expression, higher percent mitochondrial count, overall lower UMI counts).

Fig. R2- 6 Open-ST analysis of an adjacent section of the one on which we performed HCR. From left to right: Number of UMIs per spatial spot, number of UMIs per spatial spot but with an added outline to mark the ventricle and reduced opacity of the plot for better visibility, percentage of mitochondrial counts per spatial spot, myl7 expression per spot.

3) Fig. S10a-e: better data should be provided to validate the transgenes, since both are newly created in this manuscript. In particular, whether the mCherry signals in the Tam-treated double transgenic embryo represent macrophages remains unclear. At the very least, higher magnification examples including brightfield should be shown to prove that it's not autofluorescence of pigment cells that is being detected in the RFP channel. Better would be co-stainings with macrophage markers.

We thank the reviewer for the comment. We have now added images of *Tg(mpeg1.1-2A-creERT2,gcry1:NLS-EGFP)^{as602};Tg(hsp70l:loxp-TagBFP-loxp-DN-MyD88-t2A-mCherry)^{bn5711};Tg(mpeg1:eGFP)* treated with Tamoxifen which show that mCherry is co-expressed with eGFP from the mpeg1 reporter line and therefore macrophage-specific. (Fig. R2- 7)

Fig. R2- 7 Representative fluorescent images of 5 dpf *TgKI(mpeg1.1-2A-CreERT2;gcry1:NLS-eGFP)^{as602}; Tg(hsp70l:loxp-TagBFP-loxp-DN-MyD88-t2A-mCherry)^{bn5711}; Tg(mpeg1:eGFP)^{9l22}* larvae treated with TAM, showing mCherry expression in mpeg1⁺ macrophages, indicating successful Cre-mediated recombination. Chromatic aberration alignment was performed on overlaid images. Scale bar: 500 μ m. **Figure now included as Supplementary Fig. 14g**

4) Fig.S10g-h: also these images are not convincing. High magnification images should be presented to prove that the presumed increase in mCherry signals is seen in cells that at least by location could be macrophages. Better would be a co-staining with macrophage markers, e.g. the mfap4.1. HCR done in Fig. 4.

We now provide immunostainings on heart sections from 3 dpci cryoinjured *Tg(mpeg1.1-2A-creERT2,gcry1:NLS-EGFP)^{as602};Tg(hsp70l:loxp-TagBFP-loxp-DN-MyD88-t2A-mCherry)^{bns711}* ventricles treated with Tamoxifen and co-stained with dsRed and Mfap4 antibodies to show high spatial resolution images of the recombined cells in adult zebrafish after cryoinjury. This staining shows co-expression of Mfap4 and mCherry in the injured area where macrophages are recruited (Fig. R2- 8).

Fig. R2- 8 Representative images of Mfap4 (macrophage, yellow) and mCherry (recombined signals, magenta) immunostaining in heart sections from 3 dpci cryoinjured *TgKI(mpeg1.1-2A-CreERT2;gcry1:NLS-eGFP)^{as602};Tg(hsp70l:loxp-TagBFP-loxp-DN-MyD88-t2A-mCherry)^{bns711}* zebrafish treated with TAM. White dashed lines outline the injured region. Scale bar: 100 μ m. **Now included as Supplementary Fig. 15c.**

5) Can the authors show that macrophage-specific overexpression of the dnMyD88 actually reduces the pro-inflammatory phenotype of macrophages in the heart? They use il1b expression mainly as marker for that cell state, both in the (single cell) sequencing data and the HCR in situ. qPCR on whole ventricles does not show a difference, maybe HCR does? Or can they perform a scRNASeq experiment on fish expressing the construct?

As the reviewer correctly points out, the qPCR was performed on whole ventricles. In the qPCR experiment, a potential reduction of il1b expression in macrophages could have been masked by unchanged levels in other, more abundant, cell types. We therefore performed immunostainings of vehicle and Tamoxifen treated *Tg(mpeg1.1-2A-creERT2,gcry1:NLS-EGFP)^{as602};Tg(hsp70l:loxp-TagBFP-loxp-DN-MyD88-t2A-mCherry)^{bns711}* zebrafish and performed immunostainings with Mfap4 and Il1b antibodies. Quantification showed a significant reduction in Mfap4+Il1b+ cell density and overall Mfap4+ cell density in Tamoxifen treated zebrafish in the injury area at 3 and 7 dpci. These experiments indicate that DN-MyD88 expression reduces the number of pro-inflammatory macrophages in the injury area. (Fig. R2-9)

Fig. R2- 9 e, Representative images of Il1b (magenta) and Mfap4 (yellow) immunostaining of sections from cryoinjured ventricles from *TgKI(mpeg1.1-2A-CreERT2;gcry1:NLS-eGFP)^{as602};Tg(hsp70l:loxp-TagBFP-loxp-DN-MyD88-t2A-mCherry)^{bns711}* zebrafish treated with vehicle or TAM at 3 and 7 dpci. White dashed lines outline the injured area. Scale bar: 100 μ m. **f**, Magnified views of white-boxed region in (e). Arrows point to pro-inflammatory macrophages (Il1b⁺ Mfap4⁺) in the injured area. Scale bar: 10 μ m. **g**, Quantification of Mfap4⁺ cell density and Il1b⁺ Mfap4⁺ cell density in the injured area; n = 10 vehicle hearts and n = 8 TAM treated hearts at 3 dpci; n = 8 vehicle control hearts and n = 7 TAM treated hearts at 7 dpci. Dots represent individual ventricles; data shown as mean \pm s.d. Statistics: Student's t-test. **Now included as Fig. 5e-g.**

6) Fig. 5e, g, h: images are too small and individual channels (higher magnification views) need to be shown to allow readers to validate expression patterns and co-expression of markers. in e it's unclear how the wound area was identified (presumably by DAPI pattern, which is however not shown).

We have now increased all displayed image sizes of immunofluorescent stainings and included zoom-in panels for better visibility of molecule co-expression. The injury area was identified with DAPI staining patterns and the tissue structures. (Fig. R2- 10)

In panel 5f (coronary revascularization, now Fig. 6b), autofluorescence background can be found in the healthy cardiomyocytes. The injured area is a dark space without autofluorescence, which is how we identified the injured area. We have now added this information to the according figure legend.

Fig. R2- 10 Layout of images formerly shown in Fig. 5. Now displayed larger and with included zoom-in panels. Now included as Fig. 6.

7) Fig. 5h: n=4 for the 7 dpi Tam treated fish is too small.

We thank the reviewer for this important comment. We have now increased the sample sizes for the following experiments: coronary endothelial cell proliferation at 7 dpici (control n=13, TAM n=12), cardiomyocyte proliferation at 7 dpici (control n=13, TAM n=12) and cardiomyocyte dedifferentiation at 7 dpici (control n=13, TAM n=12), now displayed in **Fig. 6** (see Fig. R2- 10). With these, we obtained similar results, supporting our previous conclusions, and further strengthening our observation.

8) Fig.5i: n= 4 for the Tam-treated fish is too small, in particular considering the high variations observed in control treated fish.

We have now increased the sample sizes for the following experiments: AFOG staining at 14 dpci (control n=10 and TAM n=8, now displayed in **Fig. 6** (see Fig. R2- 10). With these, we obtained similar results, supporting our previous conclusions, and further strengthening our observation.

9) Supplementary Data Files: use of these should be improved by adding explanations to the Excel files of what the abbreviations mean and what data they actually contain. E.g. the text refers to Supplementary Data file 4 as evidence for the statement “This response may also partially arise from ambient RNA contamination from dead cardiomyocytes, which are generally the most fragile cell type to heart dissociation”. It’s entirely unclear what data this file actually contains, in particular for the “summary” sheet and how the data there support this statement...

We are now providing detailed descriptions of the content of the Supplementary Data files and explanation of the abbreviations used.

With “This response may also partially arise from ambient RNA contamination from dead cardiomyocytes [...]” we were referring to the generic response which we observe across all cell types. While this could be a true biological response that all cell types share, we wanted to add the caveat that upregulation of GO terms like Ribosome, for example, could also stem from ambient RNA contamination of dying cardiomyocytes. Supplementary Table 4 contains the list with all GO terms detected for all cell types and shows that GO terms related to the generic stress response are upregulated across all cell types.

Minor:

1) Fig. 1D. why do the dots have different colors? What does “log₁p” mean?

The dots have a slightly transparent hue which can show areas in the plot that have lower densities of dots vs higher density of dots (for example in the bottom left corner of the plot).

Log₁p is the natural logarithm of 1 plus the input value. While this abbreviation is relatively commonly used in our field, we agree with the reviewer that it should be properly introduced. We have updated the figure legend accordingly.

2) Fig. S1a: why does the x-axis contain numbers smaller than 0? The labeling is somewhat ambiguous: are the red-labelled data points remaining after applying the SNV filter or are they removed?

We apologize for the confusion. Log₁₀(CPM) stands for the log₁₀ of the counts per million. If the CPM is between 0 and 1, the log₁₀ will yield negative values. We now added a note about this in the figure legend.

The red labeled data points are those that remain after SNV filtering. As the SNV filter also applies a read coverage filter for robust SNV removal, a lot of the very lowly expressed genes are removed. As published in Neuschulz et al. we removed positions with a coverage of less than 10 reads, as well as positions that showed >25% of T-to-C conversion (likely SNVs). We now specify this more clearly in the figure legend.

3) Fig. 1E: x-Axis label missing (hours)

We thank the reviewer for the note. We have now added the x-axis label to Fig. 1e.

4) Fig. 2d. What is the difference between these data and the ones shown in Supp 2b? Figures legends indicate it should be the same data, but the conversion rates are quite different...

Thank you for the comment, we agree this was confusing. Fig. 2d shows the distribution of T-to-C conversion rates over all genes, separated per cell type. Supplementary Fig. 2b shows the distribution of T-to-C conversion rates over all cells, separated per cell type. These distributions look different mostly because highly expressed genes tend to have lower decay rates – highly expressed genes dominate the plot in Fig. S2b, but not the one in Fig. 2d where each gene contributes equally. We now added a comment about this in the legend of Fig. S2b.

5) Please rephrase “We therefore plotted the percentage of the transcriptome that is used by the expression of groups of genes”, which is confusing. The figure legend is clearer...

We thank the reviewer for the remark. We have now phrased this sentence more comprehensively.

6) Fig. 5: p-values should only be shown to significant figures. Considering the high variations observed in the data, reporting p-value to 4 decimal places does not make sense.

We thank the reviewer for the remark. We have now reduced the display of p-values to two decimals, or three decimals in case the third decimal is the first non-zero value. For transparency, we would prefer to keep the display of non-significant p-values.

Reviewer #3 (Remarks to the Author):

The manuscript by Mintcheva et al. presents a comprehensive dataset of adult zebrafish hearts after cryoinjury, combining bulk SLAM-seq, in vivo scSLAM-seq, and a large scRNA-seq atlas. The central biological claim, that a subset of macrophages is among the earliest responders, engaging Toll-like receptor (TLR) signaling and a pro-inflammatory program, is supported by sequencing readouts and orthogonal validation. The macrophage-specific, inducible MyD88 manipulation is intriguing and, if fully substantiated, would provide a persuasive functional link between very early innate sensing and regenerative outcomes. (I am not a domain expert in zebrafish cardiology; my comments focus on quantitative methods, statistics, and integrative genomics.) The study has strong potential, but several core claims currently rely on pathway-level concordance rather than gene-level integration, and key kinetic and labeling assumptions need to be demonstrated quantitatively.

We thank the reviewer for the detailed, quantitative critique focused on robustness, statistical clarity, and gene-level integration across modalities. In response, we added extensive new analyses to (i) benchmark the bulk SLAM-seq half-life estimates against published compendia and connect half-life behavior to early injury-responsive programs, (ii) quantify potential transcriptional effects of 4sU/IT injection using total-RNA analyses, and (iii) complement pathway enrichment with gene-level visualizations, effect-size reporting, and gene-signature/module-scoring across cell types and datasets. We also expanded the methodological description and validation of key modeling assumptions (including additional diagnostics/simulations and clearer reporting of filtering/identifiability considerations) and revised the manuscript/supplement accordingly to make the integrative logic and quantitative claims more transparent.

Major comments

- The list of mRNA half-lives from SLAM-seq data is a major result but its robustness remains unclear. Please benchmark against prior half-life compendia (e.g., ortholog/homolog comparisons where available). Connect these half-lives to downstream biology in this study. For instance, do short-lived transcripts disproportionately contribute to the early response programs observed by scSLAM-seq/scRNA-seq? Without integration, the half-life table reads as a detached resource.

This is a great suggestion. To benchmark our inferred decay rates in zebrafish against published mammalian half-life compendia, we assembled the curated human and mouse datasets from Agarwal et al. (2022) [1] (>60 independent experimental datasets across ActD-chase, 4sU/BrU metabolic labeling, and related methods). For each dataset, we standardized the reported half-lives (z-scored within dataset) and computed cross-dataset correlations, reproducing the clustering pattern reported by the original authors (Figure 1 in [1]). (Fig. R3-1)

[1] Agarwal, V., & Kelley, D. R. (2022). The genetic and biochemical determinants of mRNA degradation rates in mammals. *Genome biology*, 23(1), 245.

Spearman correlation across Agarwal 2022 mRNA half-life datasets

Fig. R3- 1 Spearman correlation across Agarwal 2022 mRNA half-life datasets.

We next mapped zebrafish genes to their human and mouse orthologs using Ensembl gene identifiers obtained through the BioMart query interface (Ensembl Genes 115), and computed, for each mammalian dataset, the Spearman correlation between our zebrafish decay rates and the dataset’s corresponding orthologous half-lives. We obtained 1,921 zebrafish genes with at least one human ortholog half-life measurement and 1,941 zebrafish genes with at least one mouse ortholog measurement (corresponding to 1,967 human and 2,037 mouse distinct orthologs). Across datasets, the zebrafish–mammal correlations ranged from approximately -0.20 to 0.45 , with no strong organization by method or cell type (Fig. R3- 2). This magnitude and heterogeneity match the variability observed within the compendium itself, reflecting both biological differences in mRNA stability across species and cellular contexts and the substantial methodological diversity of published half-life assays.

Fig. R3- 2 a, Spearman correlations (ρ) between zebrafish gene-level half-life z-scores and published half-life z-scores from Agarwal & Kelley 2022 across human datasets. Each point represents one dataset (computed across orthologous genes quantified in both zebrafish and the respective human dataset) and datasets are ordered by correlation value. Points are colored by experimental labeling or transcriptional inhibition method (left; ActD, 4sU, BrU4sU, BrU, Aman or 5EU) or by originating cell type (right; HEK293, HeLa, K562, RPE, H1ESC, A549, MCF7, HepG2, GM lymphoblastoid lines, B cell). The vertical grey line indicates $\rho = 0$. **b**, Same analysis as in a for mouse datasets, with points colored by experimental method (left) or by cell type (right; mESC, 3T3, MEF, EB, Neuro2a, Dendritic, C2C12 or M2-10B4). **Now included as Supplementary Fig. 2a,b**.

To assess global similarity, we performed a joint principal component analysis (PCA) across all Agarwal et al. datasets and included our zebrafish estimates as an additional condition. We constructed a gene \times dataset matrix of standardized half-lives, retained genes measured in $\geq 50\%$ of datasets, and imputed remaining entries using per-gene means to preserve stability patterns. After scaling genes to zero mean and unit variance, PCA revealed no strong clustering by method, species, or cell type, consistent with the diffuse correlation structure noted above. Importantly, the zebrafish dataset did not appear as an outlier, indicating that our inferred decay rates fall squarely within the established biological and methodological variability of existing half-life compendia. (Fig. R3- 3)

Fig. R3- 3 c, Principal component analysis (PCA) of gene-level half-life z-scores across human + zebrafish datasets. A gene \times dataset matrix of orthologous genes was constructed, retaining genes quantified in $\geq 50\%$ of

the datasets and imputing remaining missing values with the gene-wise mean. Datasets, including the zebrafish decay vector, were z-scored and projected into PCA space; points are colored by experimental method (left) or by cell type (right). The zebrafish dataset from this study is highlighted with a star; axes indicate PC1 and PC2 with the corresponding explained variance. **d**, Same PCA analysis as in **c** for mouse + zebrafish datasets, with points colored by experimental method (left) or by cell type (right). **Now included as Supplementary Fig. 2 c,d.**

As suggested by the reviewer, we next investigated the correlation of half-lives and injury response: We analyzed the distribution of half-lives of genes that were identified as injury-responsive in macrophage-like cells with scSLAM-seq, and we compared this distribution to the half-lives of non-injury-responsive genes in macrophage-like cells. For this analysis, we performed the following filtering steps:

- we only used genes with an $R^2 > 0.7$, i.e. a good fit for the half-life calculation
- we only used genes that are expressed in macrophage-like cells
- we only used genes with inferred half-lives below 48 h, i.e. a biologically expected half-life

We then performed a Wilcoxon rank sum test which yielded a p-value of 0.01. While this p-value is technically “significant” and we observe a slightly higher median half-life among non-responsive genes compared to injury responsive genes, the differences between the distributions do not appear major (Fig. R3- 4). We hence conclude that short-lived as well as long-lived transcripts contribute to the injury response. This is an important finding, since changes in transcription rate of long-lived transcripts are harder to measure over short timescales, so the benefit of scSLAM-seq is more immediate for these genes. We now added this analysis in Supplementary Fig. 5 and included a short discussion in the manuscript.

Fig. R3- 4 Half-lives of injury-responsive versus expressed, but not injury-responsive genes in macrophage-like cells with a fit $R^2 > 0.7$. Now included as Supplementary Fig. 5d.

The existence of injury-response genes with long half-lives prompted us to take a closer look at potential explanations. While injury-responsive genes might be expected to have below-average half-lives, our scSLAM-seq-based injury response also yielded metabolic pathways (OXPHOS and Glycolysis / Gluconeogenesis) and an enrichment of ribosomal genes. Especially ribosome-related genes tend to have a high expression in zebrafish heart data (both scRNA-seq and scSLAM-seq). We therefore investigated whether there are pathway-specific differences in half-lives. To this end, we used all genes assigned to a pathway (with an $R^2 > 0.7$), irrespective of it they were detected as injury-responsive or not. This analysis showed that among the three visualized pathways, the innate immune response pathways show the lowest half-lives (we did not detect any genes with $R^2 > 0.7$ in the Glycolysis / Gluconeogenesis pathway). This analysis could indicate that innate immune signaling pathways have shorter half-lives than other pathways. We have added Fig. R3- 5 in Supplementary Fig. 5 and briefly discuss it in the main text.

Fig. R3- 5 Half-lives of all genes of the OXPHOS, Ribosome and pro-inflammatory (TLR, NLR and CLR) KEGG pathways (irrespective of scSLAM-seq injury response) with a fit $R^2 > 0.7$. **Now included as Supplementary Fig. 5e.**

- Prior reports show that 4sU at similar exposure windows can perturb transcriptional profiles in mammalian systems. Please quantify whether 4sU injection alters total-RNA expression: PCA/UMAP of uninjected vs injected controls (your Fig. 1b cohort), and differential expression of total RNA for each time point vs 0 h (using the dataset underlying Fig. 1c). This is essential to rule out 4sU-induced inflammation-like artifacts.

We thank the reviewer for this important comment. We agree that both 4sU as well as intrathoracic (IT) injections can elicit a transcriptional response. To assess the magnitude of these effects, we performed PCA analysis of the time course experiments. We observe that 0h (uninjected), 0.5h and 1h timepoints closely localize in PCA space, but that there is a marked separation from later time points. (Fig. R3- 6d)

Next, we performed differential gene expression on the total RNA (disregarding labeling information) to test for transcriptional effects induced by IT injection of 4sU. When comparing the 6h time point with uninjected hearts, we observe upregulation of immune-related genes, for example *mmp9*, *mmp13a*, *cxcl18b*, *lect12* and *irf1b*, confirming that either 4sU or IT injections trigger an immune response. (Fig. R3- 6e)

Fig. R3- 6 **d**, PCA of bulk SLAM-seq time course data based on total, i.e. labeled + unlabeled counts. **e**, Volcano plot of differential gene expression of 6 h vs. 0 h (uninjected) time course datapoints. **Now included in Supplementary Fig. 1d,e.**

We further performed DGE of all other timepoints compared to uninjected (0h) hearts confirming that 4sU triggers a transcriptional response (Fig. R3- 7):

Fig. R3- 7 Volcano plot of differential gene expression compared to 0 h (uninjected) time course datapoints.

To further disentangle the effects of 4sU vs the effects of IT injections, we performed PCA analysis of uninjected hearts, IT injected zebrafish hearts (6h labeling), IP injected zebrafish hearts (6h labeling), IT injected zebrafish + sham (7h labeling, sampled at 6hps), and IT injected zebrafish + cryoinjury (7h labeling, sampled at 6hpci). This analysis showed that the effect of sham and cryoinjury are much larger as measured by separation in PCA space than the effect of just the 4sU injection (Fig. R3- 8). However, we wish to note that the libraries of IP and IT+sham/cryoinjury samples were prepared by a different experimenter and with a slightly different library setup, so we cannot exclude a potential batch effect. In summary, while injection of 4sU leads to an inflammation-like reaction, the magnitude of this response is much smaller than the reaction to cryoinjury,

Fig. R3- 8 PCA of un.injected, IT-injected 6 h, IT injected 6 hps (sham), IT-injected 6 hpci (cryoinjury) and IP injected 6 h samples based on total, i.e. labeled and unlabeled, reads. **Now included as Supplementary Fig. 1e.**

We would further like to add that we took the potential effects of both IT injection and 4sU into account when planning our scSLAM-seq experiments. By performing IT injection of 4sU in both the injured and sham condition at steady state, and sampling hearts both at 6h post sham/injury / 7h of labeling we can control for transcriptional effects induced by IT injection of 4sU. Moreover, we pooled 5 zebrafish hearts for each sample, thereby further reducing biological variability. Hence, the transcriptional effects of 4sU injection can indeed reduce signal-to-noise, but should not lead to a stronger distortion of the results.

- Key evidence for the statement that "Macrophages are among the earliest responders" from the scSLAM-seq data currently hinges on KEGG term enrichments (e.g., Fig. 2f). Pathway significance can be influenced by power, gene set size, and background choice. Please (i) visualize the specific differentially expressed ("early-response") genes in macrophage-like cells, (ii) show their regulation across other cell types, and (iii) provide effect sizes, not only P-values. In addition consider gene-set scores (e.g., TLR/pro-inflammatory signatures) rather than single genes for these analyses.

i & ii)

We thank the reviewer for the comment, and we now provide these visualizations (Fig. R3- 9, Fig. R3- 10) in the supplement. We are showing one example plot below for TLR pathway-specific genes upregulated in macrophage-like cells. For the visualization, we used the background model (in the example below in macrophage-like cells) and added the names of all genes that belonging to the TLR signaling pathway that pass our filtering conditions. For space reasons, we only provide the visualization of all upregulated pathways in macrophage-like cells, as well as the visualization of the TLR pathway in the other cell types.

Fig. R3- 9 scSLAM-seq background model plots (see mathematical supplement) with pathway-specific genes which are identified as injury-responsive in macrophage-like cells. Now included as Supplementary Fig. 4a.

Fig. R3- 10 scSLAM-seq background model plots with TLR signaling pathway-specific genes which are identified as injury-responsive in all other cell types. Now included as Supplementary Fig. 4b.

We also observe expression of TLR-related genes in other cell types. While macrophage-like cells stand out by higher expression (and by being detected in the scSLAM-seq analysis), we now also show and discuss the contributions of other cell types more extensively.

iii)

Here we are visualizing the pathway analysis taking effect sizes into account. We define the effect size as the number of genes from injury responsive gene list which are part of a pathway, over the total number of genes assigned to the pathway. This term is called “recall” in the GO term output and can also be found in the Supplementary Table 4-6 for all the other cell types and pathways. Here, for example, the TLR has a recall of 0.17 (Fig. R3- 11).

We agree with the reviewer that KEGG term enrichments come with a number of limitations. However, we believe this is still a valid approach for ensuring an unbiased analysis to drive initial hypothesis generation, which we then followed up on by performing thorough analyses on the scRNA-seq atlas to confirm our findings.

Fig. R3- 11 Upregulated KEGG terms of injury responsive genes in macrophage-like cells using SLAM information. Effect size displayed as dot size.

- The claim that GEP4 "strongly resembles" the scSLAM-seq program is based on overlapping KEGG terms. That is suggestive but not a validation. Please perform a joint analysis: (i) Build a common gene signature (e.g., the same early-response gene list) and score it in both datasets; (ii) Map scSLAM-seq cells onto the scRNA-seq reference (label transfer) and compare new/old RNA components for the same genes. This would substantiate that the same genes underlie the signal in both modalities.

We thank the reviewer for the comment. We performed label transfer and found systematic misassignment likely due to the methanol/IAA treatment that is part of the scSLAM-seq protocol, and which may increase ambient RNA levels; thus, we instead used gene-signature scoring across datasets, which directly tests concordance at the gene level. Below we describe our approach and results in detail:

(i)

We performed scoring of the intersect of upregulated genes in macrophage-like cells from the scSLAM-seq injury response analysis and the upregulated genes from the injury responsive GEP in a) the myeloid subcluster of the heart atlas, b) all cell types of the heart atlas, c) the total RNA of the 6 hpci scSLAM-seq dataset and d) the labeled RNA of the 6 hpci scSLAM-seq dataset. (Fig. R3- 12a-d) All of these analyses show a pronounced response in macrophage-like cells and neutrophils, indicating that the intersect between the two approaches indeed captures the transcriptional response to injury.

We further scored all injury-responsive genes specific to macrophage-like cells from the scSLAM-seq injury response analysis in the myeloid subcluster of the heart atlas, and we

found that these genes did not show any upregulation at 6 hpci (Fig. R3- 12e). Further investigation of the correlation of the usage of the injury responsive GEP with the intersect, all upregulated genes as well as all genes of the upregulated KEGG pathways shows that many pathways (e.g. Ribosome, OXPHOS, Glycolysis/Gluconeogenesis) show a broad and high expression and are therefore not correlating well with the injury responsive GEP (Fig. R3- 12f-k). We therefore argue in the main text that both cNMF and scSLAM-seq can show differential upregulation of pathways, which can arise due to the inherent methodological differences of both methods (e.g. cNMF also detecting sources of variation that precede cryoinjury).

Fig. R3- 12 a, Seurat module scores of the intersect of genes from injury-responsive genes identified in macrophage-like cells with scSLAM-seq and the injury-responsive GEP in the myeloid subcluster of the heart atlas. Clusters were grouped into cell types (macrophage-like cells, neutrophils, dendritic cells, eosinophils). b,

Seurat module scores of the intersect of genes in all cell types of the heart atlas. **c**, Seurat module score of the intersect of genes in the total RNA (i.e. labeled and unlabeled) of the 6 hpci scSLAM-seq sample. **d**, Seurat module score of the intersect of genes in the labeled RNA of the 6 hpci scSLAM-seq sample. **e**, Seurat module score of all injury-responsive genes in macrophage-like cells identified with scSLAM-seq in the myeloid subcluster of the heart atlas. Clusters were grouped into cell types (macrophage-like cells, neutrophils, dendritic cells, eosinophils). **f-k** Correlation of usage of the injury-responsive GEP (cNMF) and the gene intersect of scSLAM-seq injury response and the injury responsive GEP (**f**), all injury responsive genes in macrophage-like cells from the scSLAM-seq analysis (**g**), Ribosome KEGG pathway genes (**h**), Herpes simplex virus1 KEGG pathway genes (**i**), OXPHOS KEGG pathway genes (**j**), Glycolysis/Gluconeogenesis KEGG pathway genes (**k**). **a-d** now included as Supplementary Fig. 9.

ii)

To compare the dataset annotations from scSLAM-seq and scRNA-seq, we performed label transfer to assign the labels of the heart atlas to cells in the scSLAM-seq datasets. We observed that many cells are wrongly assigned as cardiomyocytes in both sham and 6hpci datasets (Fig. R3- 13, Fig. R3- 14). Further investigation showed that *myl7* expression (a common cardiomyocyte marker) is much higher in non-myocytes of the scSLAM-seq datasets compared to the scRNA-seq datasets (Fig. R3- 15, Fig. R3- 16, Fig. R3- 17). As the scSLAM-seq dissociation protocol is identical to the protocol used for the generation of the heart atlas, this effect is likely caused by the methanol fixation and overnight IAA conversion. It would not be surprising if cardiomyocytes, which are relatively fragile to dissociation given their large size and are the major source of ambient RNA in the heart atlas, react even more sensitively to the scSLAM-seq procedure compared to the conventional scRNA-seq protocol, leading to higher breaking/bursting of cardiomyocytes and higher ambient RNA contamination. This might lead to strong batch effects between scRNA-seq and scSLAM-seq datasets. Of note, such an effect has previously also been reported by Mitic et al., where scSLAM-seq was performed in adult zebrafish brains. For the same reasons, scSLAM-seq and scRNA-seq datasets were also not analyzed together in this publication.

Fig. R3- 13 Transfer of labels from the scRNA-seq heart atlas onto the 6 hps scSLAM-seq dataset.

Fig. R3- 14 Transfer of labels from the scRNA-seq heart atlas onto the 6 hpci scSLAM-seq dataset.

Fig. R3- 15 Myl7 expression in cell types of the scRNA-seq heart atlas

Fig. R3- 16 Myl7 expression in cell types of the 6 hps scSLAM-seq dataset.

Fig. R3- 17 *Myl7* expression in cell types of the 6 hpci scSLAM-seq dataset.

- The statement that ~50% of *mfap4.1+* cells co-express TLR-pathway markers at 6 hpci is difficult to interpret without detection baselines. Please provide: number of sections/animals, total *mfap4.1+* cells counted, per-section percentages with dispersion, background rates in *mfap4.1-* cells, and a statistical test. Ideally, normalize for probe efficiency (e.g., by including a housekeeping control).

To address the reviewer's comment, we now quantified number of *mfap4.1+* macrophages per section. Quantification of number of *mfap4.1+* macrophages in ventricles per mm² shows high variability of the observed number of *mfap4.1+* macrophages between individual sections of the same sample. This is not surprising, as macrophages are still sparse at this early stage and the amount of blood (through which macrophages infiltrate the heart) across sections can vary a lot (Fig. R3- 18)

Fig. R3- 18 Number of *mfap4.1+* macrophages in individual 6 hpci cryosections.

We therefore also quantified the number of *mfap4.1+* macrophages per sample rather than per section by ventricle area and highlight the fractions that show *il1b* expression or no *il1b* expression. This plot shows that in sample 1 we see approx. 50% of *mfap4.1+* cells expressing *il1b*, while in sample 2 the percentage is slightly higher (Fig. R3- 19). We have now included the quantification in Supplementary Fig. 12 and provide a clear statement of sample sizes and sections imaged in the Figure legend.

Due to various experimental confounding factors (variability of cryoinjury, low number of macrophages, section-to-section and sample-to-sample variation, blood volume per section, etc.) we believe the utility of a more detailed quantitative analysis may be relatively limited. Our main goal with this analysis was less of a quantitative statement but rather showing that

co-expression if *mfap4.1* and *il1b* exists and is not a dissociation artifact. However, the reviewer's comment made us realize that our statement “~50% of *mfap4.1*+ cells co-express [...]” was misleading, since it suggested a level of quantitative analysis that we did not aim for (and which is also not required for our biological reasoning). We therefore changed the claim from “~50%” to a “substantial fraction”.

On a qualitative level, this observation is confirmed by a new set of immunofluorescence experiments included in Supplementary Fig. 12.

Fig. R3- 19 Quantification of *mfap4.1*+ (gray) and *mfap4.1*+ *il1b*+ (pink) positive cells from HCR on 6 hpci heart cryosections. Number of sections imaged in 6 hpci #1: 6, 6 hpci #2: 3. **Now included as Supplementary Fig. 12a**

Major technical/methodological comments

- Suppl. Fig. 1a suggests considerable differences between red and blue points for lowly expressed genes, which seems more than "subtle".

As the reviewer correctly notes, we lose many lowly expressed genes if we apply SNV filtering. The reason for this is that we required a coverage of at least 10 reads per position for SNV blacklisting (see also our reply to the reviewer's following comment below). We agree with the reviewer that this difference is not "subtle", and we rephrased this sentence. Indeed, it is crucial to check if the biological effects may be caused by SNVs. We do this later in Supplementary Fig. 5, where we do find that the biological signal persists even when applying strict SNV filtering (this is what we meant by "subtle", but we agree that this was phrased in a misleading manner).

- Clarify whether blacklisting of SNVs occurs by read counts or UMI counts, and how you handle read-level T>C inconsistency within the same UMI (e.g., majority rule, consensus thresholds).

We performed blacklisting of SNVs as described in Neuschulz et al, Mol Syst Biol, 2024, and we now refer to this publication. Here is a summary of the procedure:

- 1) filter for primary reads, remove PCR duplicates (*)
- 2) pileup for every position that has a T in the reference, based on reads (no consensus for UMIs)
- 3) positions are blacklisted as potential SNVs if more than 25% of reads for that position have a C instead of T
- 4) positions are blacklisted if they have a coverage of less than 10 reads (not enough coverage to rule out an SNV)

(*) PCR duplicates are removed by flag 1024, multimappers are also removed by flag 256. Primary mapping reads (flags 0 and 16) are kept (basically: we remove everything that does not have a 0 or 16 flag). Afterwards we focus on sequencing quality for positions containing substitutions. We do not remove reads based on number of

mismatches, since reads with multiple labelling events are bound to have more mismatches due to the T to C substitutions.

Step 4 is a very strict filter that removes many positions due to low coverage. Our reasoning here is that with a coverage that low we cannot with reasonable certainty determine if a T-to-C substitution is a SNV or a labelling event. Since we still retained the biological signal after very strict SNV filtering, we concluded that SNVs did not have a major effect on the results (Supplementary Fig. 5).

The published pipeline of Neuschulz et al. does not account for read-level T>C inconsistencies for reasons of computational cost. This strategy, however, was shown to work well as shown in Neuschulz et al., Fig. 1C and Fig. EV1.

- The supplement implies degradation rate is inferred from the decay of the unlabeled fraction. This assumes near-complete replacement of unlabeled RNA for short-lived transcripts within your 12 h window. The *fosl2* fit (Fig. 1e) appears inconsistent with full disappearance. Please (i) state the identifiability conditions explicitly, (ii) show simulations probing bias when unlabeled RNA persists, and (iii) report how often modeled plateaus deviate from the assumption.

We thank the reviewer for raising this point. We have now clarified the identifiability conditions, quantified how often our data violate the full-replacement assumption, and evaluated potential bias when unlabeled RNA persists.

For each gene (N = 14,553 total genes), we quantified a data-driven observed plateau by taking the last timepoint of the time window (12 h post 4sU injection). The distribution of these plateau values displays a clear multimodal structure. To characterize it, we fitted a Gaussian mixture model to the distribution, which identified three distinct components as the best fit (Fig. R3- 20, Fig. R3- 21):

- A low-plateau component centered at ~0.11, comprising 40% of genes. These genes are poorly labeled within the 12-hour window and likely reflect background signal, transcriptional inactivity, or very long half-lives.
- An intermediate component centered at ~0.39, comprising 26% of genes. These partially labeled genes may result from time-varying transcription rates $\alpha(t)$ or intermediate kinetics (see simulations below).

Genes in these first two components are not compatible with simple degradation-kinetic models and are excluded during filtering.

- A high-plateau component centered at ~0.74, comprising 34% of genes. These correspond to well-detected, short- and medium-lived transcripts that approach a steady-state plateau near the labeling efficiency/detection limit.

This distribution is consistent with previous SLAM-seq studies, where incomplete labeling and background detection have been shown to affect plateau estimation.

Fig. R3- 20 Distribution of observed labeling plateaus across genes. For each gene ($N = 14,553$), an observed plateau was computed as the mean labeled-RNA fraction over the last part of the labeling window (12 h labeling). A Gaussian mixture model with three components was fitted to the distribution.

As expected, our simple kinetic model (which does not include a labeling correction factor and converges to a labeling ratio of 1 at long times) fits best the high-plateau component. This subset ($N = 3,167$ genes) corresponds to the high-confidence fits used in the main text.

Fig. R3- 21 Comparison of plateau distributions for all genes and for genes retained after filtering for downstream decay-rate fitting.

Importantly, observed plateau values were not correlated with total read depth (Spearman $\rho = -0.05$), indicating that this behavior is probably not driven by coverage (Fig. R3- 22).

Fig. R3- 22 Relationship between observed plateau values and mean total read depth per gene. Filtered genes used for decay-rate estimation are highlighted in blue.

To illustrate how time-varying transcription affects the approach to the labeled-RNA plateau, we simulated the dynamics of labeled L and unlabeled O RNA using a standard kinetic labeling model. One approach described for example in [2] introduces into this modeling framework a labeling-correction coefficient, denoted here as ρ . In this approach, total RNA has a synthesis rate constant α and a degradation rate constant γ . The labeled RNA has a reduced effective synthesis rate $\rho\alpha$ but the same degradation rate γ . Symmetrically, the unlabeled fraction has an effective synthesis rate $(1 - \rho)\alpha$, meaning that a fraction of newly synthesized reads is assigned to the unlabeled pool.

[2] Qiu, X., Zhang, Y., Martin-Rufino, J. D., Weng, C., Hosseinzadeh, S., Yang, D., ... & Weissman, J. S. (2022). Mapping transcriptomic vector fields of single cells. *Cell*, 185(4), 690-711.

We integrate this model under constant, increasing, and decreasing transcriptional profiles $\alpha(t)$, showing that smooth production shifts can naturally generate diverse plateau values in the labeling ratio $r_{lab}(t)$. In more detail, the system is described by:

$$\frac{dL}{dt} = \rho \cdot \alpha(t) - \gamma \cdot L(t)$$

$$\frac{dO}{dt} = (1 - \rho) \cdot \alpha(t) - \gamma \cdot O(t)$$

Where

- $\alpha(t)$ is the time-dependent transcription rate,
- γ is the degradation rate,
- ρ is the labeling correction coefficient.

In the simulations shown below, we numerically integrate this system to compute $r_{lab}(t)$ under three transcriptional regimes:

(i) constant transcription:

$$\alpha(t) = cst = 1.0,$$

(ii) Transcriptional activation, modeled as a sigmoidal increase:

$$\alpha(t) = \frac{1}{\left(1 + e^{-\frac{(t-t_0)}{k}}\right)},$$

(iii) Transcriptional repression, modeled as the corresponding sigmoidal decrease:

$$\alpha(t) = 1 - \frac{1}{\left(1 + e^{-\frac{t-t_0}{k}}\right)}$$

We used the following parameter values in the simulations presented below:

- $\rho = 0.8$ and $\gamma = \ln(2)/HL$ with $HL = 4.0$ h.
- $t_0 = 4.0$ h, the time at which activation or repression reaches its half-maximum (left panel), or a range of t_0 values in the delayed-activation experiments (right panel).
- $k = 0.8$ h, the slope parameter controlling the steepness of the sigmoidal transition.

These simulations show that smooth changes in $\alpha(t)$, generate a wide range of observable plateaus in $r_{lab}(t)$, depending on the timing of transcriptional shifts relative to the degradation timescale (Fig. R3- 23, Fig. R3- 24). This demonstrates that incomplete approach to a plateau can arise from biologically plausible transcriptional dynamics rather than from model mis-specification.

Fig. R3- 23 Simulated dynamics of the labeled RNA fraction under constant, increasing, and decreasing transcription rates $\alpha(t)$ using a kinetic labeling model with a labeling correction coefficient $\rho=0.8$.

Fig. R3- 24 Simulated dynamics of the labeled RNA fraction for increasing delays in transcriptional activation, illustrating the dependence of the observed plateau on the timing of transcriptional changes.

To assess the impact of an imperfect labeling correction on decay rate estimation, we compare the two models describing the labeling ratio:

$$r_{lab}(t) = 1 - e^{-\gamma t} \text{ and } r_{lab}(t) = \rho(1 - e^{-\tilde{\gamma}t})$$

Then, the difference between the decay rates is given by

$$\gamma - \tilde{\gamma} = -\frac{1}{t} \ln(1 - r_{lab}) + \frac{1}{t} (1 - r_{lab}/\rho) = \frac{1}{t} \ln \left(\frac{1 - r_{lab}/\rho}{1 - r_{lab}} \right)$$

Considering the function $f(r_{lab}) = \frac{1 - r_{lab}/\rho}{1 - r_{lab}}$ we note that it is strictly decreasing for $0 < \rho < 1$ and satisfying $f(0) = 1$. Therefore $\ln \left(\frac{1 - r_{lab}/\rho}{1 - r_{lab}} \right) < 0$ implying that for all times $\gamma - \tilde{\gamma} < 0$. Consequently, an misestimation of the labeling ratio leads to a systematic underestimation of the decay rate, affecting all genes in the same direction.

Similarly, the magnitude of the estimated decay rate can be compared by examining the derivative of the labeled fraction. In the two cases, $r_{lab}(t) = \gamma e^{-\gamma t}$ and $r_{lab}(t) = \rho \tilde{\gamma} e^{-\tilde{\gamma}t}$ which at initial time yield slopes γ and $\rho \tilde{\gamma}$, respectively. Since ρ is close to 1 (≈ 0.8 in our data), the difference in initial slopes remains limited, consistent with a moderate but systematic underestimation of the decay rate.

Minor (but hopefully helpful) comments

Line 110: add a reference supporting the statement about the average mRNA half-life and single-injection sufficiency.

We thank the reviewer for the comment. We now provide references of half-lives in mouse fibroblasts and mouse embryonic stem cells (<https://doi.org/10.1093/dnares/dsn030>, <https://doi.org/10.1038/nature10098>) which show that our labeling time is in the range of average half-lives.

Later in the manuscript, we indeed find that also the calculated half-lives in our data are within this range. For further clarity, we now added the following statement (based on the half-lives determined from our time course data):

We identified more than 2400 genes for which we could reliably estimate half-lives ($R^2 > 0.7$, Supplementary Table 1, Supplementary Table 2). **The average half-life amounted to 5.6 h ($R^2 > 0.7$, t[0.5h, 48h]).** Among those genes, we identify 1598 genes with a half-life shorter than 6 h (e.g. *fosl2* with $t = 0.85$ h) and 838 genes with a half-life longer than 6 h (e.g. *myh6* with $t = 8.34$ h).

Clarify "conversion rate" vs "labeling rate": are these identical? Is the denominator all genomic T positions covered, or only in reads assigned as labeled? Why does ~1% qualify as "low overall labeling rate" (L111)—relative to what benchmark?

We do not distinguish between labeling rate and conversion rate, but we use "conversion rate" in the context of quantification, and "labeling" on a conceptual level. We have now streamlined our use of these expressions in the manuscript.

For T-to-C conversions, the denominator uses all genomic T positions covered.

In our experience compared to e.g. early zebrafish embryos* and the adult zebrafish brain** 1% is a relatively low labeling rate (but still generally within the expected range).

* Holler et al., Nature Commun, 2021 (<https://doi.org/10.1038/s41467-021-23834-1>);

Tan et al, Dev Cell, 2024 (doi: 10.1016/j.devcel.2024.01.024);

Fishman et al, Nat Commun, 2024 (<https://doi.org/10.1038/s41467-024-47290-9>)

** Mitic et al., Mol Sys Bio, 2024 (doi: 10.1038/s44320-024-00022-z]

Lines 169 & 172: the core kinetic calculation resembles early SLAM-seq formulations (e.g., fold-change of labeled RNA; see Muhar et al., Science 2018). The novel contribution appears to be your noise-dominated gene filtering. Consider moving the filtering rationale/results into the main text and trimming the generic derivation, or at least reframing to highlight what is new here.

We thank the reviewer for the comment. We are now referencing the approach by Muhar et al. in the main text (and also in the Mathematical Supplement), and we now highlight our new contributions (background model) more strongly in the main text. That said, we feel that the full mathematical details of the background model would be too technical for the main text, so we would prefer to keep this part in the mathematical supplement. We hope this is in line with the reviewer's request.

Line 249: Please specify how cluster names were derived (marker-based, label transfer, manual curation).

The myeloid scRNA-seq data was clustered and marker genes calculated for each cluster using Seurat. Among the macrophage-like cell clusters, one of the top marker genes was used to define the cluster, for example *macrophage-like cells (il1b)* or *macrophage-like cells (mafb)*.

For the miloR analysis, we use the same data and UMAP embeddings as for the analysis in Fig. 3d, and therefore we use the same annotation of clusters. We now specify this in the main text.

Explain “TLR layer 1” in Figure 4e; The figure is difficult to interpret without cross-referencing Suppl. Fig. 7. Please add a schematic or brief explanation in the main figure legend about what “layer 1” denotes and how it was derived.

We thank the reviewer for the note. We have now added an explanatory sketch to Fig. 4 (Fig. R3- 25)

Fig. R3- 25 Schematic representation of dividing pathways into layers. **Now included as Fig. 4e.**

I suggest to avoid the term “simple back-of-the-envelope calculation” (L169);

We thank the reviewer for the comment, and agree with the suggestion. We have now rephrased it in the main text and the mathematical supplement.

REVIEWER COMMENTS

Reviewer #1 (Remarks to the Author):

The authors have adequately addressed my major concerns, and I have no further comments.

We would like to thank the reviewer for their positive feedback on our revised manuscript.

Reviewer #1 (Remarks on code availability):

The provided link does not work. There is no such record on GitHub.

We thank the reviewer for the comment. We have now set the GitHub repository to public.

Reviewer #2 (Remarks to the Author):

Overall, the authors have addressed all issues raised by me in a satisfactory manner. Yet, one major problem remains: they now provide higher magnification views of their immunofluorescence experiments, which allow for assessment of the validity of their conclusions. Yet, several of these raise important questions that cast doubt on the rigor with which these stainings have been analyzed. Specifically,

We thank the reviewer for their careful evaluation of the revised manuscript and for their assessment that most concerns have been satisfactorily addressed. We appreciate the remaining comments regarding the immunofluorescence images and have carefully revised the figures accordingly. Our responses are provided below.

Selected higher-magnification zoom-in panels that show signal-colocalization of immunofluorescence stainings are further provided in Fig. R2-1 (see below). We have updated Fig. 6 to show better-quality zoom-ins.

1) Fig. 6a: why does PCNA staining in several instances appear as large “blobs” that are much larger than DAPI+ nuclei (most evident in the lower right panels).

We thank the reviewer for this important observation. The larger size of some PCNA signals compared with DAPI-stained nuclei is primarily due to differences in signal detection and amplification. PCNA was detected using primary and fluorophore-conjugated secondary antibodies, which amplify the signal, whereas DAPI directly stains DNA without amplification. This difference can of course make PCNA signals appear larger or more intense relative to DAPI.

In addition, PCNA exhibits cell cycle–dependent subnuclear localization. During S phase, PCNA accumulates at replication foci, forming bright clusters that can appear as large punctates or blob-like structures. Such patterns are well-established and reflect sites of active DNA replication rather than nonspecific staining.

To further confirm staining specificity, we examined individual sections and verified that PCNA signals colocalized with DAPI-positive nuclei. Furthermore, we have now adjusted contrast scaling in the revised figure to avoid signal saturation and improve accurate visualization. These revisions clarify the presentation while the staining pattern and interpretation remain unchanged.

2) Fig. 6a: why do arrows, which are supposed to point to PCNA+ flia+ cells, in many cases clearly NOT point to such double positive cells? This is for example very evident in the upper left (control) panel.

We thank the reviewer for pointing out this important issue. We agree that in the original version of the figure, some arrows were not optimally positioned and could be misinterpreted as point to cells that were not clearly double-positive for PCNA and Flia. This problem was due to oversights during figure assembly and examination.

We have now corrected the figure to point to clearly identifiable PCNA+ Flia+ double-positive cells. Importantly, this correction does not affect the quantification shown in this figure or the conclusions of the study, as all the analyses were performed based on systematic evaluation of the full image sets rather than the representative examples shown.

3) Fig. 6c: also here, the arrows which are supposed to point out Mef2+ N2.226+ cells/nuclei partly point to very spurious examples (e.g. upper right “TAM”, right-most cell).

We thank the reviewer for pointing out this important issue. In the original version, the colored image display and signal intensity made co-localization difficult to clearly appreciate, and some arrows did not optimally point to representative Mef2+ N2.226+ double-positive cells.

To address this problem, we have now revised the figure by presenting the amplified images in grayscale, which improves contrast and allows weak signals to be more clearly visualized. In addition, we have carefully re-evaluated and repositioned the arrows to point to clear and unambiguous double-positive nuclei. These revisions improve the clarity and accuracy of the figure without affecting the quantification or conclusions.

4) Fig. 6c: also here some PCNA+ nuclei are pointed out, where no DAPI staining seems to exist, e.g. lower left panel

In the original images, weak DAPI signals were difficult to visualize due to display scaling and color representation, which may have created the impression that some PCNA+ nuclei lacked corresponding DAPI staining.

To improve visualization, we have now converted the amplified images to grayscale and adjusted contrast levels. In the revised figure, the corresponding DAPI-positive nuclei are now clearly visible. We also re-examined and corrected arrow placement where necessary

to ensure accurate representation. These changes improve visualization but do not alter the underlying data analysis or conclusions.

5) Image quality throughout is an issue. Most channels are way too overexposed (everything except DAPI).

We thank the reviewer for highlighting this issue. In the original images, contrast was increased to visualize weak signals, which resulted in this apparent overexposed appearance. In the revised figures, we have adjusted contrast levels and presented single channels in grayscale to preserve signal-to-noise information while avoiding saturation. These changes substantially improve image clarity without affecting quantification or interpretation.

Rigorous quality control would 1) not overexpose channels to an extent where all subtleties of signal/noise assessment are lost 2) would exclude PCNA stains that do not overlap with DAPI from analysis, and 3) obviously would not count cells as double positive if they are not...

We appreciate the reviewer's critical point about rigorous image analysis. In response, we have implemented the following improvements in the revised figures:

1. Single-channel images were converted to grayscale to enhance signal visibility while preserving signal-to-noise assessment.
2. Contrast levels were reduced to avoid overexposure and retain signal subtleties.
3. Arrows were repositioned to point to clear and unambiguous double-positive nuclei.

Importantly, all quantitative analyses were originally performed using strict criteria, including exclusion of PCNA signals not overlapping with DAPI and counting only bona fide double-positive cells. The revised figures more accurately reflect these rigorous analytical standards.

Fig. R2- 1 High magnification views of immunofluorescence images displayed in Fig. 6. a, Magnified views of representative cells displayed in Fig. 6a. Red arrows point to proliferating cECs ($Fli1\alpha^+ PCNA^+$). **b,** Magnified views of representative cells displayed in Fig. 6c. Red arrows point to de-differentiating cardiomyocytes ($Mef2^+ N2.261^+$) within the border zone. **c,** Magnified views of representative cells displayed in Fig. 6d. Red arrows point to proliferating cardiomyocytes ($Mef2^+ PCNA^+$) within the border zone. Scale bars: 10 μm . **Now updated in Fig. 6**

Reviewer #3 (Remarks to the Author):

I would like to congratulate the authors on a thorough and comprehensive revision. However, one significant concern remains regarding the appropriateness of the kinetic model: The new analyses clearly show that labeled RNA plateaus at ~0.8 instead of 1 (which is an inherent assumption of their kinetic model). Consequently, decay rates are systematically underestimated. Based on theoretical considerations regarding the difference of expectations of the decay rates, the authors postulate that this underestimation is "moderate". However, the implications of such bias remain unclear. Specifically, while the ranking of decay rates seems to be unaffected in expectation, how does the misspecification behave in the presence of noise? To what extent are R^2 values inflated which are used for filtering, potentially biasing results based on decay rates?

I do not want the authors to provide a more in-depth theoretical analysis. Rather, it should be straight-forward to estimate the labeling correction parameter directly from the data and include it into the model. Presenting model fits that utterly deviate from data (as in Figure 1e) detracts from the high quality of this otherwise very strong manuscript, and I strongly recommend incorporating this correction.

We would like to thank the reviewer for their positive feedback on our revised manuscript. We appreciate the reviewer's comment about the kinetic model, which we now addressed in the following manner: We have now updated all analyses related to the RNA half-lives to contain the plateau-corrected half-lives. Accordingly, we have updated Fig. 1e, Supplementary Fig. 2, Supplementary Fig. 5d,e, Supplementary Table 2, as well as the mathematical supplement. This correction led to a higher number of genes with R^2 values above 0.7 and an overall shorter average half-life which we updated in the main text.